# Estrogenic-dependent glutamatergic neurotransmission from kisspeptin neurons governs feeding circuits in females

Jian Qiu[1], Heidi M Rivera[1], Martha A Bosch[1], Stephanie L Padilla[2], Todd L Stincic[1], Richard D Palmiter[2], Martin J Kelly[1,3], Oline K Rønnekleiv[1,3]*

[1]Department of Physiology and Pharmacology, Oregon Health and Science University, Portland, United States; [2]Department of Biochemistry, Howard Hughes Medical Institute, University of Washington, Seattle, United States; [3]Division of Neuroscience, Oregon National Primate Research Center, Oregon Health and Science University, Beaverton, United States

**Abstract** The neuropeptides tachykinin2 (Tac2) and kisspeptin (Kiss1) in hypothalamic arcuate nucleus Kiss1 (Kiss1[ARH]) neurons are essential for pulsatile release of GnRH and reproduction. Since 17β-estradiol (E2) decreases *Kiss1 and Tac2* mRNA expression in Kiss1[ARH] neurons, the role of Kiss1[ARH] neurons during E2-driven anorexigenic states and their coordination of POMC and NPY/AgRP feeding circuits have been largely ignored. Presently, we show that E2 augmented the excitability of Kiss1[ARH] neurons by amplifying *Cacna1g, Hcn1 and Hcn2* mRNA expression and T-type calcium and h-currents. E2 increased *Slc17a6* mRNA expression and glutamatergic synaptic input to arcuate neurons, which excited POMC and inhibited NPY/AgRP neurons via metabotropic receptors. Deleting *Slc17a6* in Kiss1 neurons eliminated glutamate release and led to conditioned place preference for sucrose in E2-treated KO female mice. Therefore, the E2-driven increase in Kiss1 neuronal excitability and glutamate neurotransmission may play a key role in governing the motivational drive for palatable food in females.
DOI: https://doi.org/10.7554/eLife.35656.001

*For correspondence:
ronnekle@ohsu.edu

## Introduction

Hypothalamic arcuate nucleus kisspeptin (Kiss1[ARH]) neurons and anteroventral periventricular and periventricular nucleus Kiss1 (Kiss1[AVPV/PeN]) neurons are regulated in a species-, sex, and gonadal steroid-specific manner (*Estrada et al., 2006*; *Lehman et al., 2013*; *Ramaswamy et al., 2008*; *Smith, 2008 Smith et al., 2006*). Kiss1[AVPV/PeN] neurons are essential for positive feedback by estradiol on gonadotropin-releasing hormone (GnRH) neuronal activity and the luteinizing hormone (LH) surge in rodents (*Clarkson et al., 2008*). 17β-estradiol (E2) increases *Kiss1* mRNA expression in these rostral Kiss1 neurons (*Smith et al., 2005*), and their excitability is increased by the E2-driven upregulation of T-type calcium, hyperpolarization-activated, cyclic-nucleotide gated (h)- and persistent sodium ($I_{NaP}$) currents (*Piet et al., 2013*; *Wang et al., 2016*; *Zhang et al., 2015,2013*). In addition, glutamate induces burst firing and pacemaking activity in Kiss1[AVPV/PeN] neurons (*Wang et al., 2017*; *Zhang et al., 2013*).

In contrast to the Kiss1[AVPV/PeN] neurons, E2 significantly inhibits *Kiss1* mRNA expression in ARH neurons (*Lehman et al., 2010*; *Navarro et al., 2009*; *Smith et al., 2005*). However, the effect of E2 on ion channel expression and excitability is less clear in Kiss1[ARH] neurons, and we do not fully understand the mechanism(s) by which E2 affects the excitability of the Kiss1[ARH] neuronal population

(*Zhang et al., 2015*). Kiss1[ARH] neurons co-express the peptide neurotransmitters kisspeptin (Kiss1), tachykinin 2 (Tac2; *a.k.a.* neurokinin B; NKB) and dynorphin (*Lehman et al., 2013*), which have been proposed to be responsible for pulsatile release of GnRH and LH (*Clarkson et al., 2017*; *Li et al., 2009*; *Navarro et al., 2009*). Indeed, the cellular mechanisms by which Kiss1[ARH] neurons synchronize to drive GnRH pulses was recently elucidated and involves a combination of co-released Tac2 excitation via Tac3 receptors and dynorphin presynaptic inhibition via κ-opioid receptors (*Qiu et al., 2016*). Kiss1[ARH] neurons also express the vesicular glutamate transporter 2 (vGluT2) (*Cravo et al., 2011*; *Nestor et al., 2016*), an indication that they have the potential to package and release the neurotransmitter glutamate (*Herman et al., 2014*). Optogenetic activation of Kiss1[ARH] neurons evokes glutamatergic EPSCs not only in Kiss1[AVPV/PeN] neurons (*Qiu et al., 2016*) but also in proopiomelanocortin (POMC) and neuropeptide Y/agouti-related peptide (NPY/AgRP) neurons (*Nestor et al., 2016*), which further establishes direct functional connections between Kiss1[ARH] neurons and hypothalamic neurons important for the control of reproduction and energy homeostasis. In males, *Slc17a6* (encodes vGluT2) mRNA and glutamate release are increased in gonadectomized as compared to intact animals, an indication of inhibitory effects of gonadal steroids on glutamate neurotransmission in these neurons (*Nestor et al., 2016*). Based on pronounced male/female differences in feeding behavior and the excitatory glutamatergic input from Kiss1[ARH] neurons to Kiss1[AVPV/PeN] neurons in females (*Asarian and Geary, 2006*; *Qiu et al., 2016*), we hypothesized that the mRNA expression of *Slc17a6* and glutamate release in females in contrast to males, would be amplified by E2 in Kiss1[ARH] neurons in order to help maintain reproduction during different energy states. Therefore, we investigated the effects of ovariectomy (OVX) and E2-replacement on neuronal excitability as well as the effect of E2 on mRNA expression of *Slc17a6* in Kiss1[ARH] neurons and the release of glutamate from these neurons onto Kiss1[AVPV/PeN], POMC and NPY/AgRP neurons. Also, to evaluate the main functional role of glutamate release from Kiss1[ARH] neurons in females, we deleted the expression of *Slc17a6* in Kiss1 neurons and measured electrophysiological changes in Kiss1[ARH] neuronal transmission as well as behavioral changes in vivo.

## Results

### Estradiol treatment reduces the mRNA expression of neuropeptides while increasing the expression of the T-type calcium channels, HCN channels, and the excitability of Kiss1[ARH] neurons

It has been shown repeatedly in a number of species, that the mRNA expression of the peptide neurotransmitters kisspeptin, Tac2 and dynorphin, which are co-expressed in Kiss1[ARH] neurons, is increased in OVX animals and reduced in E2-treated animals (*Lehman et al., 2013*). However, vGluT2 is also expressed in Kiss1[ARH] neurons (*Cravo et al., 2011*; *Nestor et al., 2016*), and *Slc17a6* mRNA levels are reduced in intact as compared to castrated males, an indication of sex-steroid regulation (*Nestor et al., 2016*). Currently, we used a sensitive real-time qPCR assay to measure relative quantitative differences of *Kiss1, Tac2, Pdyn, Tacr3, Slc17a6* and ion channels in manually-harvested Kiss1[ARH] neurons in OVX as compared to E2-treated females (*Figures 1–3*). Based on the CT values (*Figure 1*) and quantitative analysis with *Tacr3* as calibrator (see Materials and Methods), *Tac2* mRNA was the most highly expressed in Kiss1[ARH] neurons from OVX females, such that *Tac2* >>> (50-fold) *Pdyn* ≥ *Kiss1* >> (6-fold) *Slc17a6*, and with E2 treatment, *Tac2* >> (18-fold) *Pdyn* >> (2-fold) *Slc17a6* > (2-fold) *Kiss1* (one-way ANOVA). These results suggested that there could be a dramatic change in the Kiss1[ARH] neuronal synaptic transmission to various target neurons (see below for further quantitative and functional analyses).

Previously, we found that an E2-treatment that induces an LH surge in OVX females increases mRNA expression and/or function of a number of ion channels important for neuronal excitability, including T-type calcium channel subunit $Ca_V3.1$, sodium channel subunits $Na_V 1.1\alpha$ and $Na_V \beta2$ and the corresponding currents $I_T$, $I_{NaP}$ and $I_h$ in Kiss1[AVPV/PeN] neurons (*Zhang et al., 2015, 2013*). Since the channels that underlie the pacemaking activity of Kiss1[AVPV/PeN] neurons ($Na_V 1.1\alpha$, $Na_V \beta2$ mRNA and $I_{NaP}$) were not altered by E2-treatment in Kiss1[ARH] neurons (*Zhang et al., 2015*), we hypothesized that E2 similarly would not alter *Cacna1g* mRNA (encoding Cav3.1), *Hcn* mRNA expression or the associated T-type calcium- and h- currents, respectively, in Kiss1[ARH] neurons. Surprisingly, we found that the mRNA expression of *Cacna1g* and *Hcn1* and *Hcn2* were significantly increased in

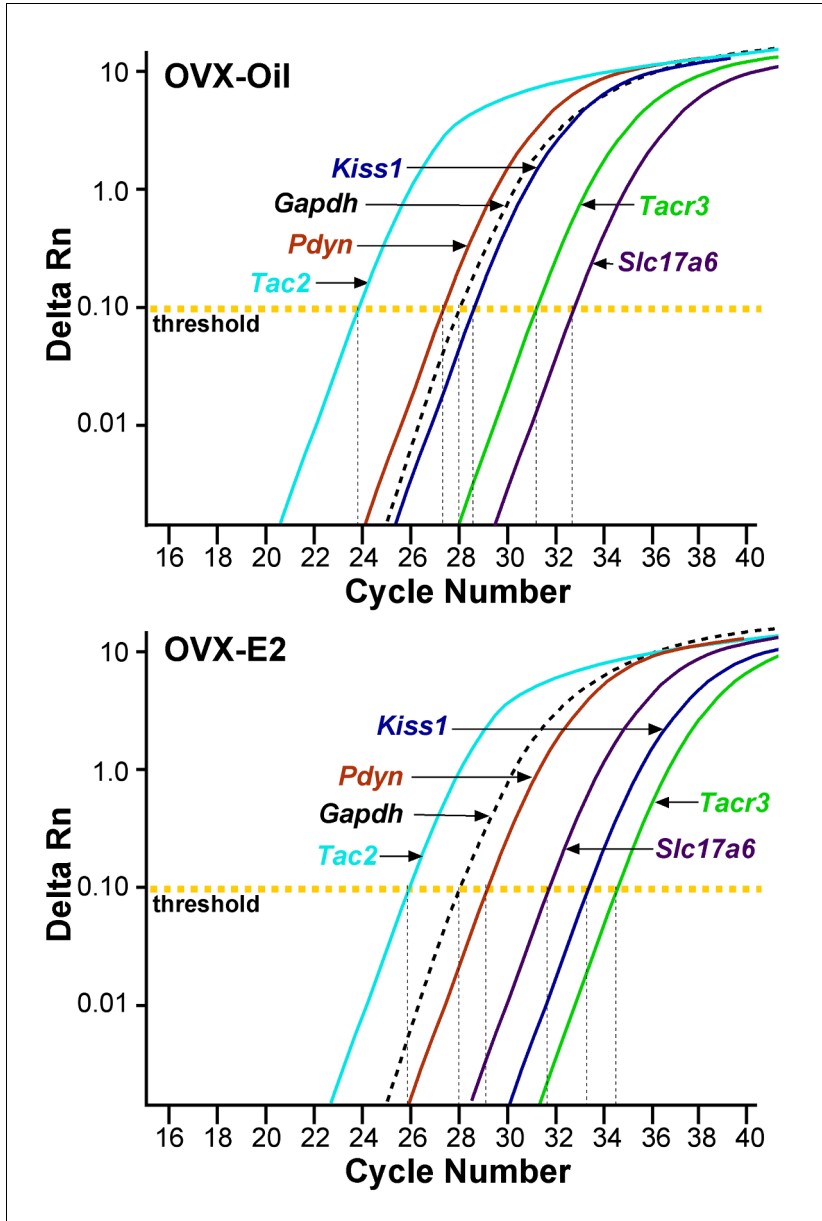

**Figure 1.** qPCR amplification assay illustrating the cycle threshold (CT) for the different neuropeptides and vGluT2 in Kiss1[ARH] neurons. Cycle number is plotted against the normalized fluorescence intensity (ΔRN) to visualize the PCR amplification of *Tac2, Pdyn, Kiss1, Tacr3, Slc17a6* and the reference gene *Gapdh* in 5 cell Kiss1[ARH] pools obtained from oil- and E2-treated, OVX animals. The amplification efficiency for each primer pair is listed in *Table 1*. These efficiencies allowed us to use the comparative ΔΔCT methods for quantification. The cycle threshold (CT; horizontal dashed line) is the point in the amplification from which sample values were calculated using the $2^{-\Delta\Delta CT}$ equation as described in the Methods.

DOI: https://doi.org/10.7554/eLife.35656.002

The following source data is available for figure 1:

**Source data 1.** OVX + Oil Gene comparisons (fold change generated using OVX + Oil Tacr3 as calibrator) and OVX + E2 Gene comparisons.

DOI: https://doi.org/10.7554/eLife.35656.003

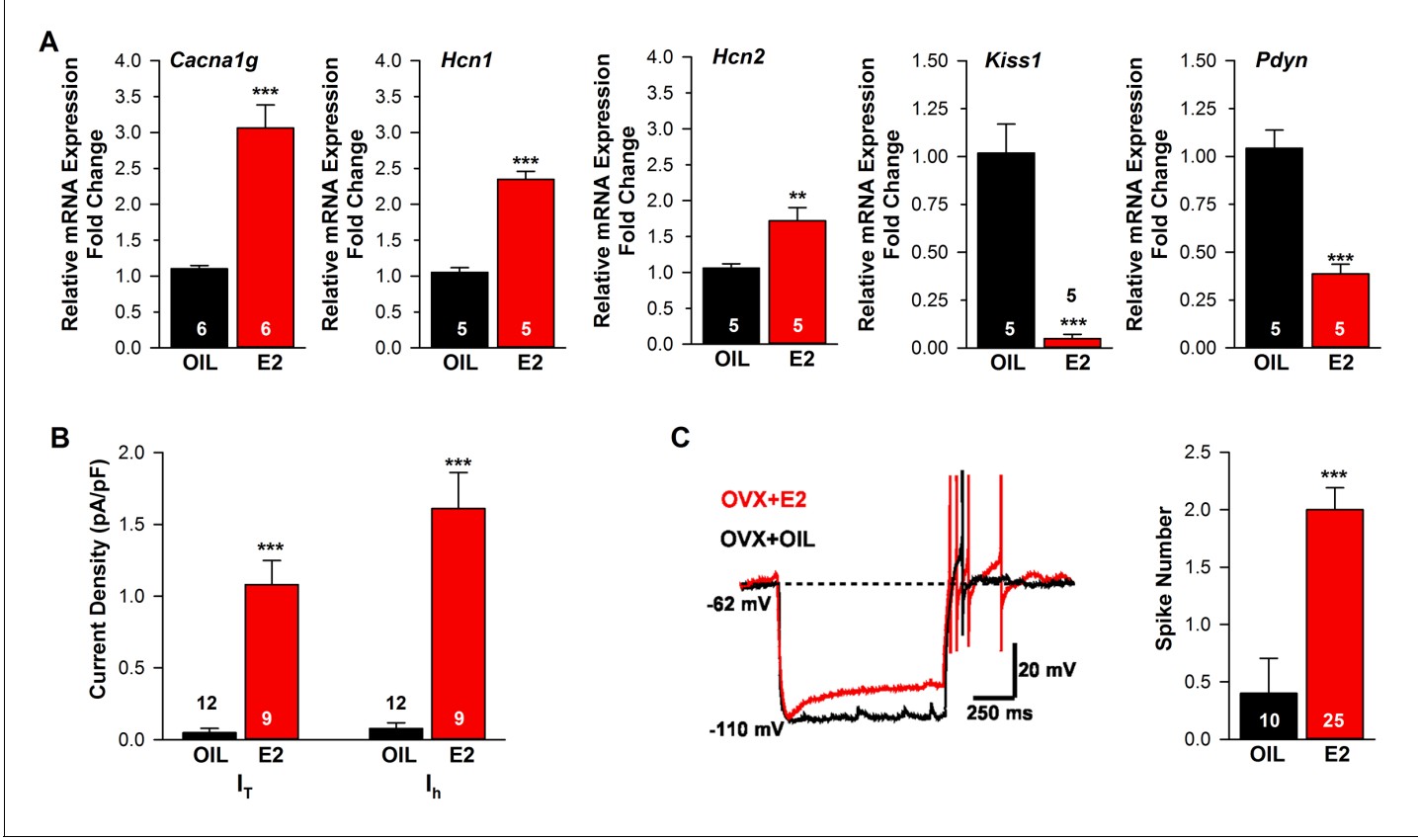

**Figure 2.** Estradiol regulation of ion channel mRNA expression and excitability of Kiss1[ARH] Neurons. (**A**) Quantitative real-time PCR measurements of *Cacna1g* (Cav3.1), *HCN1*, *HCN2*, *Kiss1* and *Pdyn* mRNAs in Kiss1[ARH] neuronal pools (3 pools of 5 cells each per animal) from OVX oil- and E2-treated mice (n = 5–6 animals per group). Note that E2 increased the mRNA expression of *Cacna1g, Hcn1, Hcn2,* but as expected decreased the mRNA expression of *Kiss1* and *Pdyn* in the same Kiss1 neuronal pools (for, *Cacna1g*, Unpaired t-test $t_{(10)}$ = 6.037, p<0.001; *Hcn1*, Unpaired t-test, $t_{(8)}$ = 10.13, p<0.0001; *Hcn2*, Unpaired t-test, $t_{(8)}$ = 3.420, p<0.01; *Kiss1*, Unpaired t-test, $t_{(8)}$ = 6.348, p<0.001; *Pdyn*, Unpaired t-test, $t_{(8)}$ = 6.118, p<0.001). (**B**) T-type calcium current and h-current density (pA/pF) in Kiss1[ARH] neurons from OVX oil- and E2-treated mice (for T-current, $t_{(19)}$ = 6.956, p<0.0001; for h-current, $t_{(19)}$ = 6.964, p<0.0001; n = 9–12 neurons from 8 animals). Current densities were measured as previously described (*Zhang et al., 2013*). (**C**) Example of rebound burst firing in Kiss1[ARH] neurons (left), which increased fast Na$^+$ spiking with E2, and summary data (right) from oil- versus E2-treated females (n = 10 and 25 neurons, respectively). Rebound firing was measured as previously described (*Zhang et al., 2013*). Bar graphs represent the mean ±SEM, (Unpaired t-test, $t_{(33)}$ = 4.455, p<0.0001). **p<0.01, ***p<0.001.

DOI: https://doi.org/10.7554/eLife.35656.004

The following source data is available for figure 2:

**Source data 1.** *Cacna1g, Hcn1, Hcn2, Kiss1* and *Pdyn* expression in Kiss1-ARH neurons (*Figure 2A*).

DOI: https://doi.org/10.7554/eLife.35656.005

Kiss1[ARH] neurons from E2-treated females as compared to oil-treated, OVX females (*Figure 2A*). For comparison we also measured *Kiss1* and *Pdyn* mRNAs in the same neuronal pools, and as predicted, *Kiss1* and *Pdyn* mRNA levels were significantly reduced by E2 (*Figure 2A*). In addition, both T- and h-currents (*Figure 2B*) as well as neuronal excitability (measured as rebound excitation; *Figure 2C*) were increased in Kiss1[ARH] neurons with E2-treatment as compared to oil-treatment. Therefore, although E2-treatment downregulates neuropeptide expression in Kiss1[ARH] neurons, it significantly increases the intrinsic conductances and hence the excitability of these vital neurons.

## The mRNA for vGluT2 is up-regulated by E2-treatment in Kiss1[ARH] neurons

Having found previously that Kiss1[ARH] neurons express *Slc17a6* (encodes vGluT2) and release glutamate onto POA neurons and ARH POMC and NPY/AgRP neurons (*Nestor et al., 2016*; *Qiu et al., 2016*), we hypothesized that E2 would increase glutamatergic input to Kiss1[AVPV/PeN] neurons to

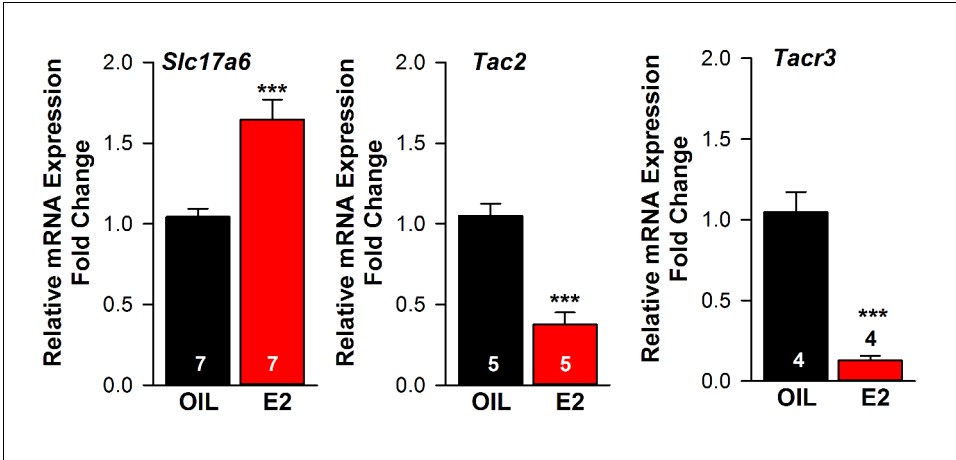

**Figure 3.** Estradiol regulation of *Slc17a6* mRNA expression in Kiss1[ARH] neurons. Quantitative real-time PCR measurements of *Slc17a6*, *Tac2* and *Tacr3* mRNAs in Kiss1[ARH] neuronal pools (3–6 pools of 5 cells each per animal) from OVX oil- and E2-treated mice (n = 4–7 animals per group). Note that E2 increased the mRNA expression of *Slc17a6*, but as expected decreased the mRNA expression of *Tac2* and *Tacr3* in the same Kiss1 neuronal pools. Bar graphs represent the mean ±SEM (for *Slc17a6*, Unpaired t-test, $t_{(8)}$ = 4.522, p<0.001; *Tac2*, Unpaired t-test, $t_{(8)}$ = 6.350, p<0.001; *Tacr3*, Unpaired t-test, $t_{(6)}$ = 7.161, p<0.001). ***p<0.001.
DOI: https://doi.org/10.7554/eLife.35656.006

The following source data is available for figure 3:

**Source data 1.** *Slc17a6, Tac2, Tacr3* expression in Kiss1-ARH neurons.
DOI: https://doi.org/10.7554/eLife.35656.007

positively influence fertility and glutamatergic input onto NPY and POMC neurons to affect feeding behavior in females. Therefore, we performed quantitative analysis of *Slc17a6*, and *Slc32a1* (encodes vGAT) in oil- and E2-treated, OVX females to explore potential E2 regulation. The qPCR analysis revealed that Kiss1[ARH] neurons (5 cells/pool; 3–6 pools/animal) in E2-treated, OVX female mice expressed approximately 2-fold higher levels of *Slc17a6* mRNA than oil-treated, OVX mice (*Figure 3*), while the level of *Slc32a1* mRNA was undetectable in Kiss1[ARH] neurons from both groups of female mice (data not shown). For comparison we also measured *Tac2* and *Tacr3* mRNAs in the same Kiss1[ARH] neuronal pools, and as expected, the mRNA expression of these transcripts were both significantly reduced in E2-treated females (*Figure 3*). Therefore, the E2-induced increased *Slc17a6* mRNA expression in Kiss1[ARH] neurons is an indication that the amino acid neurotransmitter glutamate is regulated differently by E2 than the neuropeptides in females.

## Glutamate, not GABA, is released from Kiss1[ARH] neurons

Using whole-cell, voltage-clamp recordings in slices from OVX female mice, we initially examined evoked (photostimulated) PSCs in ARH NPY/AgRP neurons from *Kiss1[Cre:GFP]::Npy[GFP]* mice or in POMC neurons from *Kiss1[Cre:GFP]::Pomc[EGFP]* mice that had received a bilateral injection of AAV1-ChR2-mCherry (or ChR2-YFP) into the ARH (*Figure 4A,B*). Subsequently, we performed blind whole-cell patch recordings from POMC and NPY/AgRP neurons in *Kiss1[Cre:GFP]* mice that had received a bilateral injection of AAV1-ChR2-mCherry (or ChR2-YFP) into the ARH. For the latter group we could segregate the medially located, higher input resistance (1.6 ± 0.2 GΩ), lower capacitance (~18.4 ± 1.0 pF) NPY/AgRP neurons from the more laterally located, lower input resistance (~1.2 ± 0.2 GΩ), higher capacitance (~24.4 ± 1.3 pF) POMC neurons (*Qiu et al., 2006*; *Smith et al., 2013*). In total, seventy-seven cells showed an inward current following photostimulation with a mean amplitude of 29.8 ± 3.4 pA, a mean latency to peak of 4.0 ± 0.1 ms, and an average decay time constant of 5.5 ± 0.6 ms. This light-induced response was blocked with the AMPA- and NMDA-receptor blockers, CNQX and AP5, respectively (*Figure 4C*). Therefore, glutamate, and not GABA, is the primary amino acid neurotransmitter released from Kiss1[ARH] neurons in females. Moreover, although TTX (1 µM) abrogated the photostimulated post-synaptic inward current, we could rescue the light-induced response with the addition of the potassium channel blocker 4-AP (100 µM to the

**Table 1.** Primer Table

| Gene name (encodes for) | Accession Number | Primer Location (nt) | Product Length (bp) | Annealing Temp (°C) | Efficiency Slope | Efficiency r$^2$ | Efficiency % |
|---|---|---|---|---|---|---|---|
| Kiss1 (Kiss1)[a,b] | NM_178260 | 64–80 167–183 | 120 | 57[a], 60[b] | −3.410 | 0.989 | 97 |
| Pomc (POMC)[a] | NM_008895 | 145–164 327–344 | 200 | 60.5 | | | |
| Npy (NPY)[a] | NM_023456 | 106–125 268–287 | 182 | 60 | | | |
| Grm1 (mGluR1)[a] | NM_001114333 | 2044–2063 2210–2229 | 186 | 59 | | | |
| Grm2 (mGluR2)[a] | NM_001160353 | 2448–2466 2574–2592 | 145 | 59 | | | |
| Grm5 (mGluR5)[a] | NM_01143834 | 1436–1453 1663–1682 | 247 | 59 | | | |
| Grm7 (mGluR7)[a,b] | NM_001346640 | 1354–1373 1445–1462 | 109 | 55[a], 60[b] | −3.306 | 0.985 | 100 |
| Npffr1 (Npffr1)[a] | NM_001177511 | 360–378 450–470 | 111 | 55 | | | |
| Kiss1r (GPR54)[a] | NM_053244 | 1900–1917 2125–2144 | 245 | 60 | | | |
| Slc17a6 (vGluT2)[c] | NM_080853 | 1038–1056 1213–1231 | 194 | 57 | | | |
| Slc17a6 (vGluT2)[b] | NM_080853 | 872–889 967–984 | 113 | 60 | −3.293 | 0.920 | 100 |
| Slc32a1 (vGAT)[b] | NM_009508 | 813–834 928–949 | 137 | 60 | −3.290 | 0.906 | 100 |
| Pdyn (Dyn)[b] | NM_018863 | 210–228 345–363 | 154 | 60 | −3.516 | 0.990 | 93 |
| Tac2 (NKB)[a,b] | NM_009312 | 79–97 207–225 | 147 | 60 | −3.324 | 0.992 | 100 |
| Tacr3 (Tacr3)[b] | NM_021382 | 764–783 864–883 | 120 | 60 | −3.504 | 0.911 | 93 |
| Cacna1g (Cav 3.1)[b] | NM_009783 | 5004–5025 5060–5083 | 80 | 60 | −3.372 | 0.968 | 98 |
| Hcn1 (HCN1)[b] | NM_010408 | 1527–1546 1641–1662 | 136 | 60 | −3.253 | 0.958 | 100 |
| Hcn2 (HCN2)[b] | NM_008226 | 1122–1143 1199–1218 | 97 | 60 | −3.279 | 0.969 | 100 |
| Gapdh (GAPDH)[b] | NM_008084 | 689–706 764–781 | 93 | 60 | −3.352 | 0.998 | 99 |
| Actb (β-actin)[b] | NM_007393 | 446–465 535–555 | 110 | 60 | −3.465 | 0.996 | 95 |

[a]primers used for scRT-PCR.

[b]primers used for qPCR.

[c]primers used to confirm the vGluT2-KO using scRT-PCR.

DOI: https://doi.org/10.7554/eLife.35656.032

bath) in both POMC (n = 3) and NPY/AgRP (n = 2) neurons (**Figure 4D,E**), which is biophysical evidence for direct synaptic signaling from Kiss1[ARH] neurons to POMC and NPY/AgRP neurons (**Cousin and Robinson, 2000**; **Petreanu et al., 2009**). Following recording, the cytoplasm of numerous responsive cells was collected and analyzed for mRNA transcripts (*Pomc*, *Npy*, and *Kiss1*) using scRT-PCR to confirm our targeting strategy described above (**Figure 4D,E**, Insets). Based on this analysis, there was no difference in the amplitude of the evoked fast PSCs between POMC and NPY neurons (*post-hoc* identified) (25.8 ± 4.8 pA, n = 36 POMC neurons versus 33.3 ± 4.9 pA, n = 41 NPY/AgRP neurons). Therefore, it appears that when Kiss1[ARH] neurons in the female are firing at low

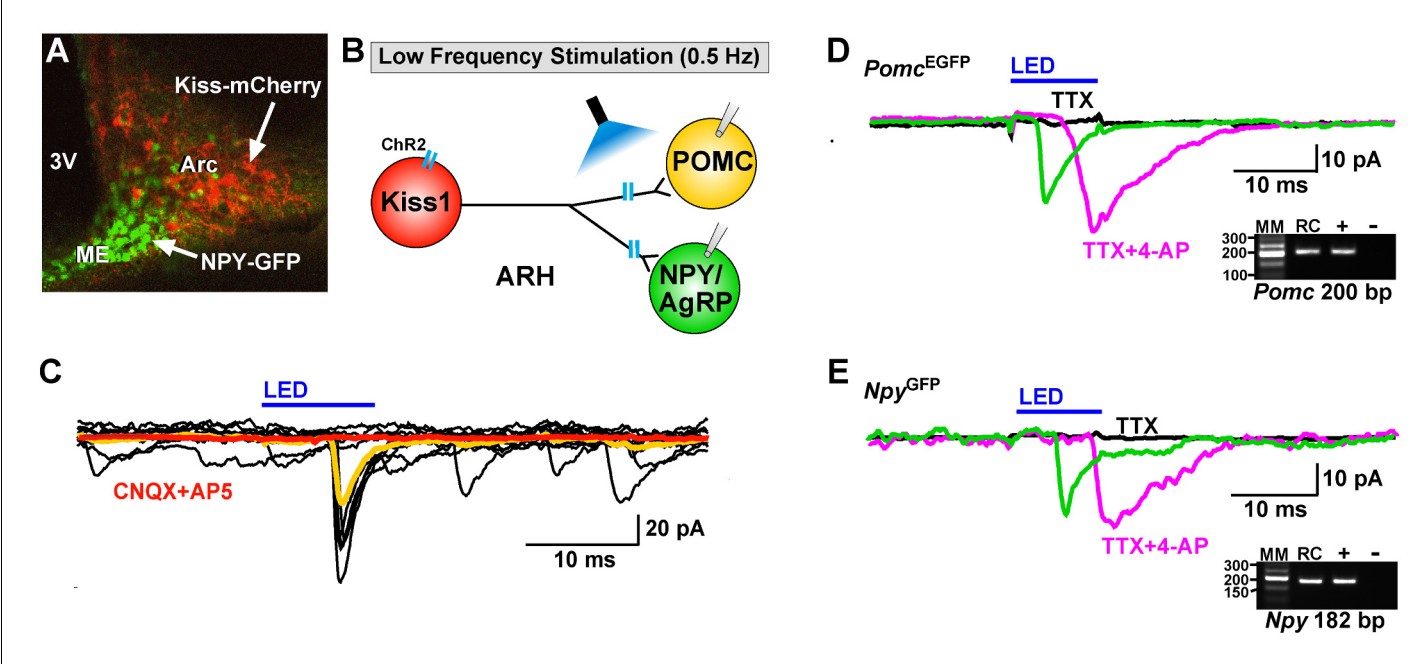

**Figure 4.** Optogenetic activation of Kiss1-ARH neurons directly excites POMC and NPY/AgRP neurons via glutamate release. (**A**) AAV1-DIO-ChR2: mCherry was bilaterally injected into ARH of Kiss1[Cre]: : Npy[GFP] mice or Kiss1[Cre]: : Pomc[EGFP] (not shown). (**B**), schematic of experimental design; whole-cell, voltage-clamp ($V_{hold}$ = −60 mV) recordings were made in POMC[EGFP] or NPY[GFP] neurons and a single pulse (intensity, 660 μW; 10 ms duration) of blue light (470 nm) was delivered to the ARH. (**C**), a fast inward current was recorded in both POMC and NPY neurons (yellow trace is average) that was antagonized by CNQX (10 μM) and AP5 (50 μM) (red trace). (**D,E**), the optogenetic (glutamate) response (green trace) was abrogated in the presence of TTX (1 μM, black trace) but rescued with the addition of the K[+] channel blocker 4-AP (100 μM, magenta trace) in both POMC (**D**) and NPY (**E**) neurons, n = 3 and 2, respectively. Insets show the scRT-PCR *post hoc* identification of representative recorded *Pomc* and *Npy* neurons. RC, recorded cell; +, positive control (with reverse transcriptase); -, negative control (without reverse transcriptase); MM, molecular marker.
DOI: https://doi.org/10.7554/eLife.35656.008

frequencies there is equivalent fast (ionotropic) glutamatergic input to both POMC and NPY/AgRP neurons in the female, similar to what we found in males (*Nestor et al., 2016*).

## Steroid-dependent glutamate release from Kiss1[ARH] neurons

E2-treatment of OVX females increased the mRNA expression of *Slc17a6* in Kiss1[ARH] neurons, and synaptic glutamate release is tightly coupled to vGluT2 expression (*Herman et al., 2014*). Therefore, we explored the differences in evoked glutamate release from Kiss1[ARH] neurons using a paired-pulse-ratio paradigm (PPR: ratio of the amplitude of the second pulse over the amplitude of the first pulse) (*Zucker and Regehr, 2002*) in control (oil) and E2-treated OVX Kiss1[Cre:GFP], Kiss1[Cre:GFP]:: Npy[GFP] and Kiss1[Cre:GFP]::Pomc[EGFP] female mice that received a bilateral injection of AAV1-DIO-ChR2:mCherry (or AAV1-DIO-ChR2:YFP) into the ARH. Using a photostimulation PPR protocol of two 5 ms LED stimulations separated by 50 ms (*Figure 5A,B*), we found that POMC and NPY/AgRP neurons in E2-treated, OVX female mice had a significantly lower PPR compared to oil-treated, OVX females in both POMC neurons (*Figure 5C,D*) and in NPY/AgRP neurons (*Figure 5E,F*). These results indicate that there is a greater probability of glutamate release with the first stimulus in E2-treated, OVX animals (*Figure 5B*), which is consistent with higher mRNA expression of *Slc17a6* in Kiss1[ARH] neurons following E2-treatment. Collectively, these data indicate that there is a striking male/female difference concerning the effects of sex steroids (testosterone versus estradiol) on *Slc17a6* mRNA expression and glutamate release in Kiss1[ARH] neurons (*Nestor et al., 2016*).

Previously, we found that Kiss1[ARH] neurons send projections to and excite Kiss1[AVPV/PeN] neurons via glutamate release (*Qiu et al., 2016*). Currently we confirmed that ChR2-mCherry injections labeled Kiss1[ARH] neurons within the arcuate nucleus only (*Figure 6A1*), and sent extensive fiber-projections rostrally to the AVPV/PeN area in the vicinity of immunoreactive Kiss1 neurons (*Figure 6A2,*

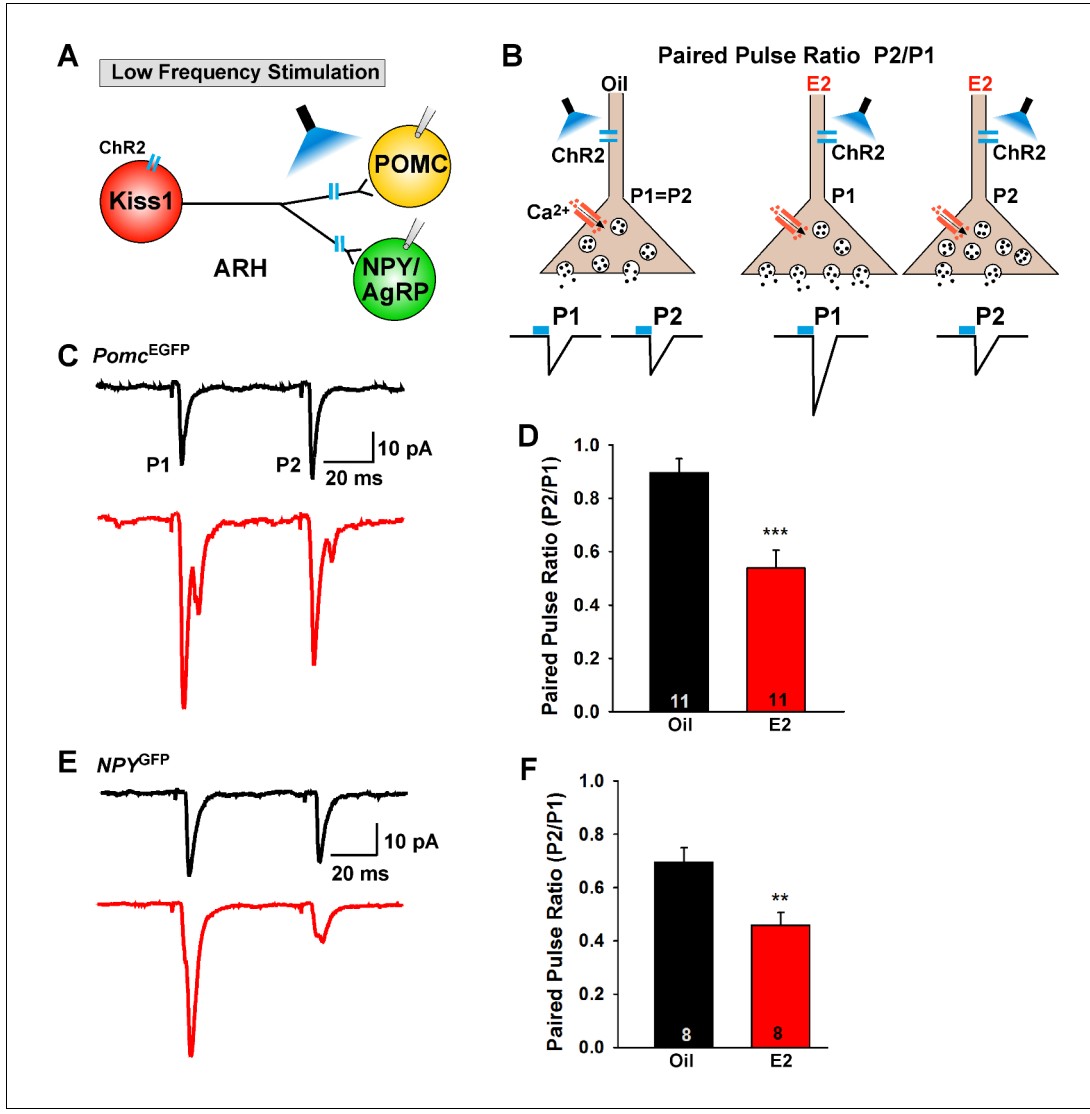

**Figure 5.** E2-treatment increases glutamate release from Kiss1[ARH] neurons onto POMC and NPY neurons. (A) schematic of photostimulation of cells/terminals of Kiss1[ARH] neurons and recording from POMC or NPY/AgRP neurons. (B), illustration of a paired-pulse regime (two blue light pulses of 5 ms duration separated by 50 ms); fast glutamatergic inward currents (P1 and P2) were recorded to measure the probability of neurotransmitter release in postsynaptic neurons. (C), AAV1-DIO-ChR2:mCherry was bilaterally injected into ARH of Kiss1[Cre]::Npy[GFP] mice or Kiss1[Cre]::Pomc[EGFP] mice. Using a paired-pulse regime, fast glutamatergic inward currents were recorded in POMC[EGFP] neurons ($V_{hold}$ = −60 mV) from both oil-treated, OVX (upper trace, black) and E2-treated, OVX (lower trace, red) females. The averaged responses (50 sweeps) are shown. (D) E2-treatment significantly decreased the paired-pulse ratio (P2/P1; indicating that there was a higher probability of glutamate release from Kiss1[Cre:GFP]-ChR2 neurons (Unpaired t-test, $t_{(20)}$ = 4.184, p<0.001). (E) similarly using a paired-pulse regime, fast glutamatergic inward currents were recorded in NPY[GFP] neurons ($V_{hold}$ = −60 mV) from both oil-treated, OVX (upper trace, black) and E2-treated, OVX (lower trace, red) females. The averaged responses (50 sweeps) are shown. (F) E2-treatment significantly decreased the paired-pulse ratio (P2/P1) indicating that there was a higher probability of glutamate release from Kiss1[Cre:GFP]-ChR2 neurons (Unpaired t-test, $t_{(14)}$ = 3.255, p<0.01). **p<0.01, ***p<0.001.

DOI: https://doi.org/10.7554/eLife.35656.009

The following source data is available for figure 5:

**Source data 1.** The paired-pulse ratio (P2/P1) for *Figure 5D and F*.
DOI: https://doi.org/10.7554/eLife.35656.010

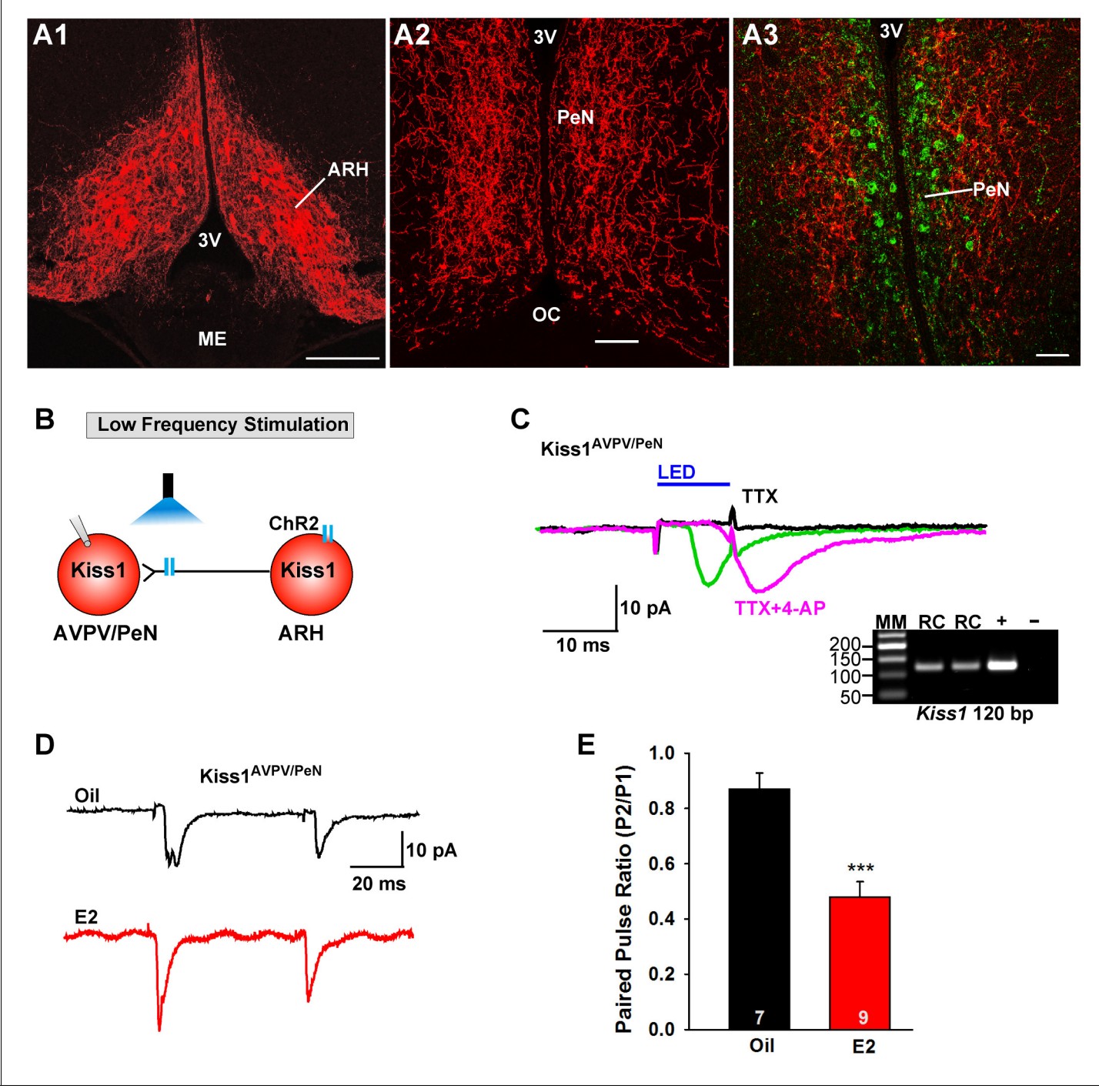

**Figure 6.** E2 treatment increases the probability of glutamate release from Kiss1[ARH] neurons onto Kiss1[AVPV/PeN] neurons. (A1-A2) Photomicrographs showing the pronounced projections of ChR2:mCherry fibers to the preoptic area including the PeN following bilateral injections of AAV1-DIO-ChR2:mCherry in the ARH of Kiss1[CreGFP] V2 mice (note that the GFP was not visible in the V2 animals). Therefore, some POA sections from the same animals were stained for kisspeptin using the Caraty 564 antibody and revealed immunoreactive Kiss1 neurons in the PeN (green cells) (A3). Essentially none of the POA somas including the Kiss1 cells expressed ChR2-mCherry. Scale bars = 100 μM (A1,A2); 50 μm (A3). (B) schematic of photostimulation of the terminals of Kiss1[ARH] neurons and recording of Kiss1[AVPV/PeN] neurons. (C) following AAV1-DIO-ChR2:YFP (or mCherry) injection into the ARH, a fast inward current was recorded in Kiss1[AVPV/PeN] neurons following blue light stimulation (green trace). The response was antagonized by CNQX (10 μM) and AP5 (50 μM) (not shown) and was abrogated in the presence of TTX (1 μM, black trace) but rescued with the addition of the K$^+$ channel blocker 4-AP (100 μM, magenta trace; n = 4 neurons). (D) using a paired-pulse regime (two blue light pulses of 5 ms duration separated by 50 ms), fast glutamatergic inward currents were recorded in Kiss1[AVPV/PeN] neurons ($V_{hold}$ = −60 mV) from both oil-treated, OVX (upper trace, black) and E2-

*Figure 6 continued on next page*

*Figure 6 continued*

treated, OVX (lower trace, red) females. The averaged responses (50 sweeps) are shown. (**E**) E2-treatment significantly decreased the paired-pulse ratio (P2/P1) indicating that there was a higher probability of glutamate release from arcuate Kiss1$^{Cre:ChR2}$ neurons (Unpaired t-test, t$_{(14)}$ = 4.748, p<0.001). ***p<0.001. Inset shows scRT-PCR *post hoc* identification of representative recorded Kiss1$^{AVPV/PeN}$ neurons. RC, recorded cell; +, positive control (with reverse transcriptase); -, negative control (without reverse transcriptase); MM, molecular marker.
DOI: https://doi.org/10.7554/eLife.35656.011

The following source data is available for figure 6:

**Source data 1.** The paired-pulse ratio (P2/P1) for *Figure 6E*.
DOI: https://doi.org/10.7554/eLife.35656.012

*A3*). Notably, ChR2-mCherry expressing cell bodies were not found in the AVPV/PeN. Thus, we determined whether this glutamatergic projection was also regulated by E2. We used a coronal slice preparation that contained only the ARH *Kiss1$^{Cre:GFP}$*-AAV1-DIO-ChR2:mCherry (or ChR2:YFP) fiber projections to the AVPV/PeN (*Figure 6A2,A3*) and explored the light-evoked glutamate release from Kiss1$^{ARH}$ neurons (*Figure 6B*). Photostimulation-induced an inward current in Kiss1$^{AVPV/PeN}$ neurons, which was blocked in the presence of TTX (1 µM) but rescued with addition of 4-AP in the presence of TTX (n = 4) (*Figure 6C*). Therefore, this is biophysical evidence for direct synaptic signaling from Kiss1$^{ARH}$ neurons to Kiss1$^{AVPV/PeN}$ neurons. Using a photoactivation-PPR protocol of two 5 ms LED stimulations as described above, we found that AVPV/PeN neurons, including Kiss1$^{AVPV/PeN}$ neurons, from E2-treated females had reduced PPR, an indication that Kiss1$^{AVPV/PeN}$ neurons, similar to POMC and NPY/AgRP neurons, receive an enhanced glutamatergic input from Kiss1$^{ARH}$ neurons in E2-treated females (*Figure 6D,E*).

## High frequency stimulation of glutamate release from Kiss1$^{ARH}$ neurons excites POMC neurons and inhibits NPY neurons via metabotropic receptors

Although optogenetic activation of Kiss1$^{ARH}$ neurons at low frequency caused similar ionotropic stimulation of both POMC and NPY/AgRP neurons, high-frequency stimulation of Kiss1$^{ARH}$ neurons/fibers, while blocking the fast AMPA/NMDA input with CNQX + AP5, had differential actions on these two different populations of ARH neurons as measured in slices from E2-treated females. High-frequency (20 Hz) stimulation of Kiss1$^{ARH}$ neurons/fibers generated a slow EPSC in POMC$^{EGFP}$ neurons, and in current clamp the stimulation depolarized POMC neurons and increased their firing rate (*Figure 7A–C*). In contrast, high-frequency optogenetic stimulation generated a slow IPSC in NPY$^{GFP}$ neurons, and in current clamp the same stimulus hyperpolarized and inhibited NPY neurons (*Figure 7A,D,E*). This high frequency response was blocked approximately 90% by the mGluR7 antagonist ADX71743 (IPSP reduced from 5.9 ± 1.2 mV to 0.6 ± 0.2 mV, n = 5; paired t-test, t$_{(4)}$ = 4.281, p=0.013). The change in membrane potential following Kiss1$^{ARH}$ stimulation was significantly different between POMC and NPY/AgRP neurons (*Figure 7F*). Therefore, high-frequency activity in Kiss1$^{ARH}$ neurons, which through presumably glutamate spillover to extrasynaptic mGluRs (*Nietz et al., 2017*; *Watanabe and Nakanishi, 2003*), excites POMC but inhibits NPY/AgRP neurons.

Next, we used scRT-PCR to measure the expression of excitatory and inhibitory metabotropic glutamate receptors in individual POMC and NPY/AgRP neurons. This analysis revealed that POMC neurons expressed group I metabotropic glutamate receptor, mGluR1 and/or mGluR5 (mGluR1, 44%, n = 5 animals; mGluR5, 21%, n = 5 animals) (*Figure 8A,B*), whereas NPY/AgRP neurons expressed primarily the group III metabotropic glutamate receptors, mGluR7, with a few neurons also expressing the group II mGluR2 (mGluR7, 50%, n = 5 animals; mGluR2, 13%; n = 4 animals) (*Figure 8C,D*). Therefore, we used the mGluR 1/5 agonist DHPG (3,5-dihydroxyphenylglycine) to explore its action in POMC neurons, and found that DHPG (50 µM) depolarized and stimulated firing in POMC neurons (*Figure 9A*), and induced an inward current in synaptically-isolated POMC neurons (*Figure 9B*). The I-V relationship for the DHPG-induced current showed a reversal at −30 mV (*Figure 9C*), indicating that a cationic current was driving the pronounced depolarization. However, there was no difference in the depolarizing effects of DHPG on POMC neurons in oil-treated as

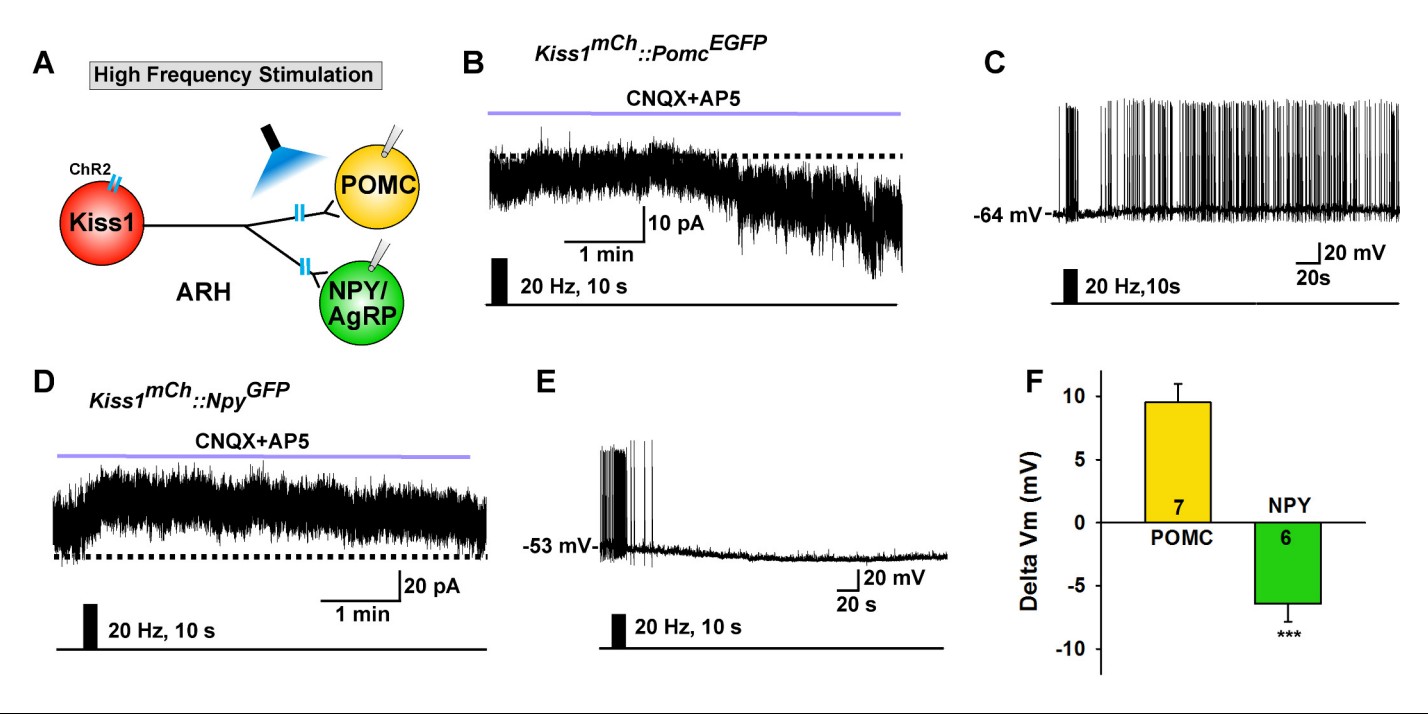

**Figure 7.** High frequency stimulation of Kiss1[Cre:GFP] neurons inhibits NPY neurons but excites POMC neurons. (A) Schematic of photostimulation of the terminals of Kiss1[ARH] neurons and recording of POMC or NPY/AgRP neurons. (B) high-frequency optogenetic stimulation (20 Hz, 10 s) of Kiss1[Cre:GFP] neurons/fibers, which were labeled with AAV-DIO-ChR2-mCherry, generated a slow EPSC in POMC[EGFP] neurons (in the presence of CNQX, 10 μM and AP5, 50 μM). (C) in current clamp the same stimulus depolarized and increased the firing frequency of POMC neurons. (D) in voltage clamp ($V_{hold}$ = −60 mV) high-frequency stimulation of Kiss1[Cre:GFP]-ChR2 neurons/fibers generated a slow IPSC in NPY[GFP] neurons (CNQX, 10 μM; AP5, 50 μM). (E) in current clamp the same stimulus hyperpolarized and inhibited the firing frequency of NPY neurons. (F) summary of the effects of high-frequency stimulation of Kiss1[Cre:GFP]-ChR2 neurons on POMC neurons (depolarized 9.5 ± 1.5 mV, n = 7) and on NPY neurons (hyperpolarized 6.4 ± 1.4 mV, n = 6). The responses (change in membrane potential, Delta Vm) were significantly different in POMC versus NPY/AgRP neurons (Unpaired t-test, $t_{(11)}$=7.685, p<0.0001). ***p<0.001.

DOI: https://doi.org/10.7554/eLife.35656.013

The following source data is available for figure 7:

**Source data 1.** Effects of high-frequency stimulation of Kiss1[Cre:GFP]-ChR2 neurons on POMC and NPY neurons (**Figure 7F**).

DOI: https://doi.org/10.7554/eLife.35656.014

compared to E2-treated, OVX females (**Figure 9D**). Confirming the scRT-PCR findings, DHPG had no effect on NPY/AgRP neurons (data not shown).

To pharmacologically elucidate the postsynaptic metabotropic glutamate response in NPY neurons, we first utilized the group II mGluR agonist DCG-IV (10 μM), and found that it hyperpolarized and inhibited firing in NPY neurons (**Figure 10A**). In addition, DCG-IV in the presence of fast sodium channel, ionotropic glutamatergic and GABAergic blockade induced an outward current in NPY/AgRP neurons (**Figure 10B**). The I-V relationship for the DCG-IV induced current exhibited inward rectification and a reversal potential at $E_K^+$ (−95 mV), the hallmark of activation of G protein-coupled inwardly rectifying $K^+$ (GIRK) channels (**Figure 10C**). Moreover, 10 μM DCG-IV was more efficacious to hyperpolarize NPY/AgRP neurons in E2-treated OVX females, indicating that there was a more robust E2-induced inhibition of NPY/AgRP neurons (**Figure 10G**). DCG IV, however, had no effect on POMC neuronal excitability (data not shown). Since mGluR7 was the most highly expressed in NPY/AgRP neurons (**Figure 8**), we also tested the mGluR7-selective agonist AMN082 (10 μM) (**Ren et al., 2011**) for its postsynaptic actions on NPY/AgRP neurons. Indeed, AMN082 hyperpolarized and inhibited firing in NPY/AgRP neurons (**Figure 10D**). It generated about a 2-fold greater outward current than DCG-IV, which reversed near $E_K^+$ (**Figure 10E,F**), and the magnitude of the hyperpolarization was significantly increased in E2-treated, OVX females (**Figure 10H**). However, AMN082 (10 μM) had no effect on a POMC neuron that exhibited a robust response to DHPG (data

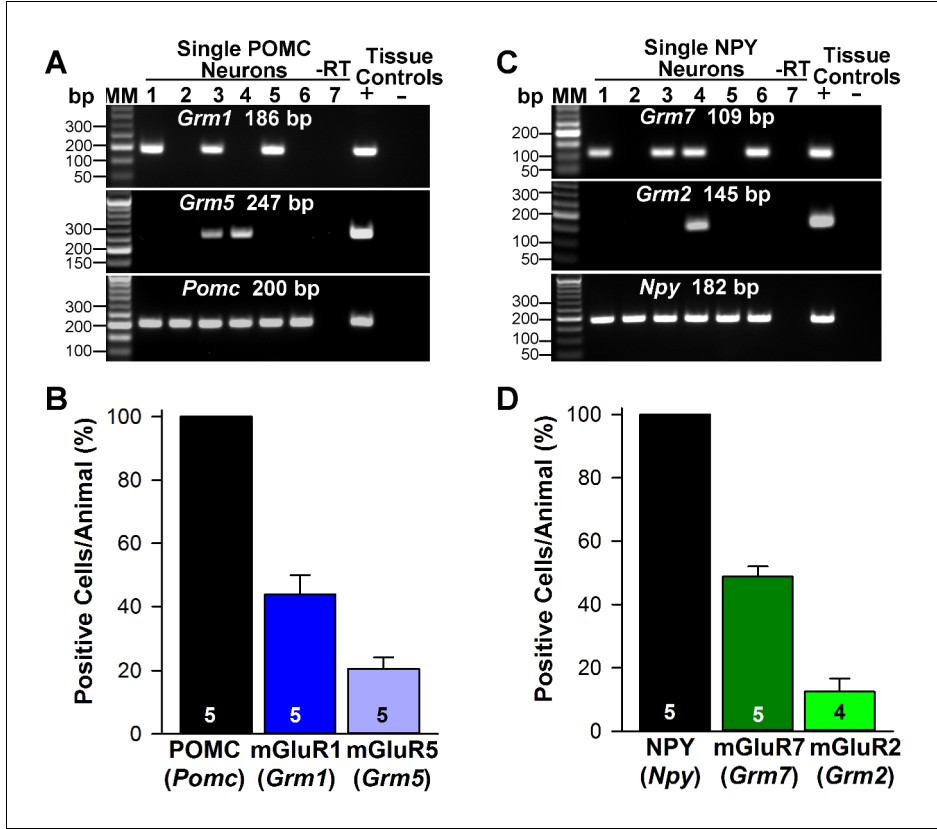

**Figure 8.** Metabotropic glutamate receptor expression in POMC and NPY neurons. (**A**) representative gels illustrating mRNA expression of *Pomc*, *Grm1* (encoding mGluR1) and *Grm5* (encoding mGluR5) in individual POMC[EGFP] neurons. The expected base pair (bp) sizes for *Pomc*, *Grm1* and *Grm5* are 200 bp, 186 bp, 247 bp, respectively. (**B**) bar graphs summarizing the percentage (mean ±SEM) of POMC[EGFP] cells (24 cells each from 5 animals) that expressed *Pomc*, *Grm1*, and *Grm5* mRNAs. (**C**) representative gels illustrating mRNA expression of NPY, *Grm2 (encoding* mGlur2) and *Grm7 (encoding* mGlur7) in individual NPY[GFP] neurons. The expected base pair (bp) sizes for *Npy*, *Grm2* and *Grm7* are 182 bp, 145 bp, 109 bp, respectively. (**A,C**) exclusion of reverse transcriptase (-RT) in a reacted cell was used as negative control. RNA extracted from medial basal hypothalamic tissue was also included as positive (+, with RT) and negative (-, without RT) tissue controls. (**D**) bar graphs summarizing the percentage (mean ± SEM) of NPY[GFP] cells (24 cells each from 5 animals) that expressed *Npy*, *Grm2*, and *Grm7* mRNAs.

DOI: https://doi.org/10.7554/eLife.35656.015

The following source data is available for figure 8:

**Source data 1.** *Pomc, Grm1 and Grm5* mRNA expression in POMC neurons (*Figure 8B*).
DOI: https://doi.org/10.7554/eLife.35656.016

not shown). In addition, the mRNA expression of mGluR7 was significantly increased in NPY neurons obtained from E2- versus oil-treated OVX females (*Figure 10I*). Therefore, E2 significantly increases the receptor expression and response of NPY/AgRP to group II/III mGluR's (Gi,o-coupled), which would contribute to the inhibition of these orexigenic neurons in heightened E2-driven reproductive states.

## Conditional knockout of vGluT2 in Kiss1[ARH] neurons eliminates glutamate release

To better understand the function of glutamate in Kiss1 neurons, we ablated vGluT2 specifically in Kiss1 neurons. Given that *Slc17a6* is not expressed in Kiss1[AVPV/PeN] neurons in our mouse model, this ablation was specific for Kiss1[ARH] neurons and, perhaps, for other Kiss1 neurons not yet shown to express *Slc17a6*, including the amygdala and bed nucleus of stria terminalis (BNST) Kiss1 neurons

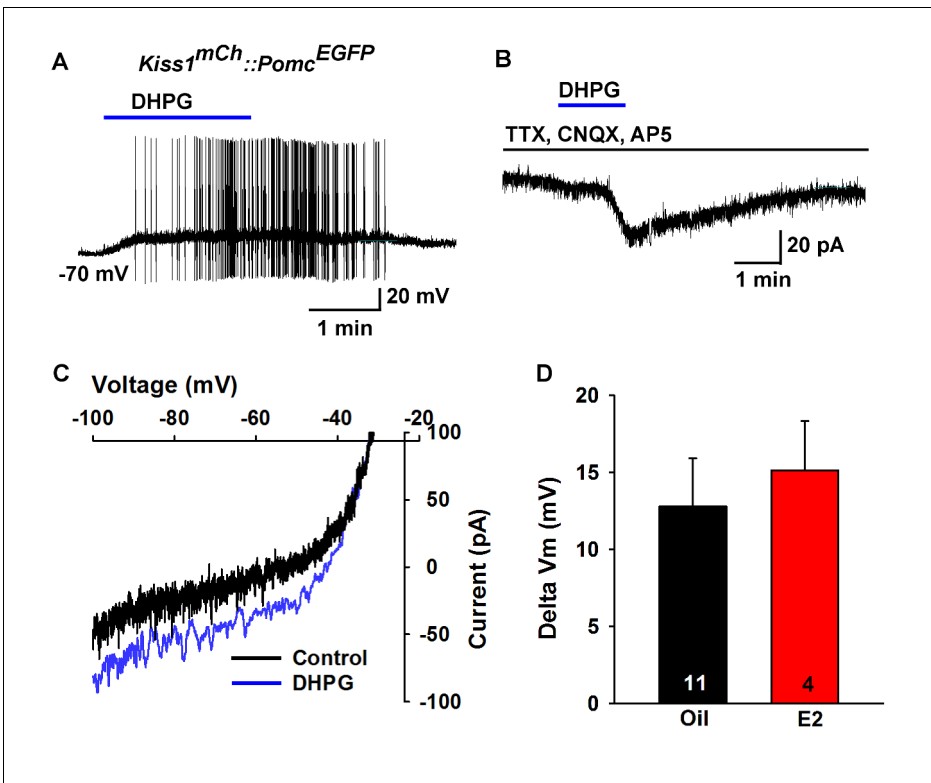

**Figure 9.** Metabotropic glutamate response is excitatory in POMC neurons. (A) metabotropic glutamate receptor 1/5 agonist DHPG (50 μM) depolarized and induced firing of a POMC neuron. (B) Rapid bath application of DHPG induced an inward current in the presence of fast sodium channel and ionotropic glutamatergic blockade, $V_{hold}$ = −60 mV. (C) voltage ramps from 0 to −100 mV were applied (over 2 s) before and during the treatment with DHPG, showed a reversal at −30 mV. (D) summary of the depolarizing effects of DHPG in POMC neurons in oil-treated and E2-treated, OVX females. There was not a significant difference in the response (Unpaired t-test, $t_{(13)}$ = 0.4168, p=0.6831).

DOI: https://doi.org/10.7554/eLife.35656.017

The following source data is available for figure 9:

**Source data 1.** Depolarizing effects of DHPG in POMC neurons in oil-treated and E2-treated, ovariectomized females (*Figure 9D*).

DOI: https://doi.org/10.7554/eLife.35656.018

(*Lehman et al., 2013*; *Qiu et al., 2016*). To delete vGlut2 from Kiss1 neurons we crossed $Slc17a6^{lox/lox}$ mice with $Slc17a6^{+/\Delta}::Kiss1^{CreGFP/+}$ mice, which yielded mice that were knockout (KO) or heterozygotes for $Slc17a6$ (Het) in Kiss1 neurons, as well as $Slc17a6$ Het and wild-type (WT) mice that lacked Cre. We confirmed the $Slc17a6$ deletion in KO mice using scRT-PCR of harvested Kiss1[ARH] neurons (*Figure 11A*). For identification of Kiss1[ARH] neurons in E2-treated animals and for functional studies, AAV1-DIO-ChR2:YFP was injected bilaterally in the ARH of $Slc17a6$ KO, $Slc17a6$ Het and control Kiss1[Cre:GFP] females. Importantly based on the scRT-PCR analysis, none of the YFP cells in KO females expressed $Slc17a6$, whereas 100% expressed $Tac2$ (*Figure 11A*). The majority of cells also expressed $Kiss1$ mRNA, although this transcript was more difficult to detect in E2-treated females (see *Figure 1*).

Based on previous findings, the slow EPSP underlying Kiss1[ARH] neuronal synchronization, is dependent on the release of Tac2 and dynorphin in Kiss1[ARH] neurons (*Qiu et al., 2016*). To explore a potential role of glutamate in this function, we generated a slow EPSP using high-frequency (20 Hz, 10 s) optogenetic stimulation in slices obtained from oil- and E2-treated, OVX females (*Qiu et al., 2016*). High-frequency optogenetic stimulation generated a similar slow EPSP amplitude in arcuate Kiss1 neurons from oil-treated, OVX Kiss1 control and oil-treated OVX $Slc17a6$ KO Kiss1 mice (*Figure 11C,D,G*). In contrast, high-frequency optogenetic stimulation of Kiss1 neurons in E2-

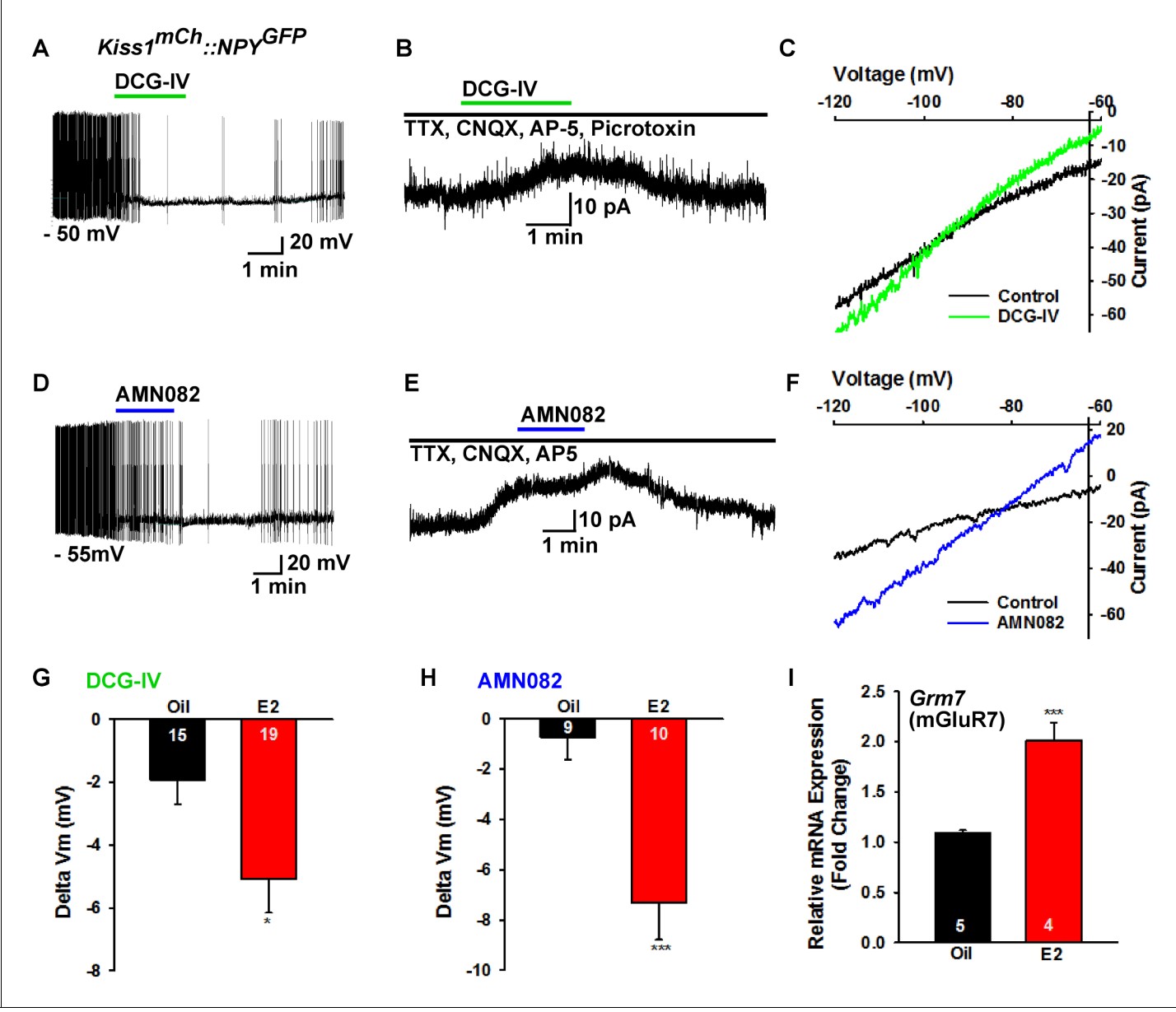

**Figure 10.** Metabotropic glutamate response is inhibitory in NPY neurons and augmented by E2. (A) the group II mGluR agonist DCG-IV (10 μM) hyperpolarized a NPY[GFP] neuron. (B) DCG-IV activated an outward current in a NPY[GFP] neuron in the presence of fast sodium channel (TTX, 0.5 μM), ionotropic glutamatergic (CNQX, 10 μM and AP5, 50 μM) and GABAergic (picrotoxin, 100 μM) blockers (V hold = −60 mV). (C) I-V relationship for DCG-IV- induced current showed a reversal at $E_{K+}$ (−95 mV). (G) DCG-IV was more efficacious to hyperpolarize NPY neurons in E2-treated versus oil-treated, OVX females (Unpaired t-test, $t_{(32)}$ = 2.261, p=0.031). *p<0.05. (D) the mGluR7 allosteric agonist AMN082 (10 μM) hyperpolarized and inhibited firing of an NPY[GFP] neuron. (E) AMN082 generated a 25 pA outward current in a NPY[GFP] neuron in the presence of fast sodium channel and ionotropic glutamatergic blockade (V hold = −60 mV). (F) I-V relationship for AMN082-induced current showed a reversal close to $E_{K+}$. (H) AMN082 was more efficacious than DCG-IV to hyperpolarize NPY neurons in E2-treated versus oil-treated, OVX females (Unpaired t-test, $t_{(17)}$ = 3.747, p=0.002). (I) Quantitative real-time PCR measurements of *Grm7* mRNA in NPY[GFP] neuronal pools (4 pools of 5 cells each per animal) from oil- and E2-treated, OVX mice (n = 4–5 animals per group). Bar graphs represent the mean ± SEM (Unpaired t-test, $t_{(7)}$=6.020, p=0.0005). ***p<0.001.
DOI: https://doi.org/10.7554/eLife.35656.019

The following source data is available for figure 10:

**Source data 1.** DCG-IV was more efficacious to hyperpolarize NPY neurons in E2-treated versus oil-treated, OVX females (*Figure 10G*).
DOI: https://doi.org/10.7554/eLife.35656.020

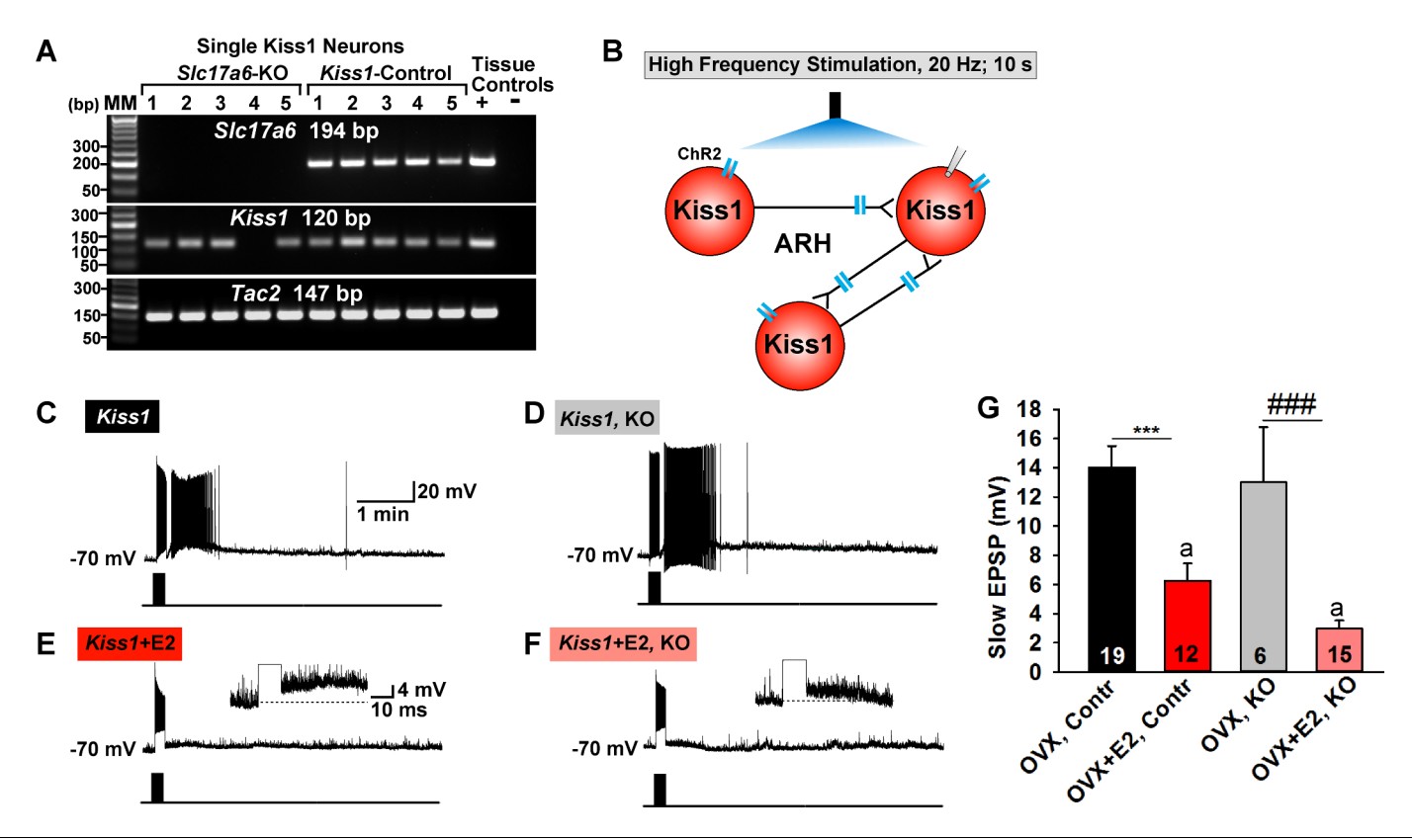

**Figure 11.** Deletion of *Slc17a6* in Kiss1ARH neurons attenuates the slow EPSP in Kiss1ARH neurons from E2-treated mice. (**A**) representative gels illustrating mRNA expression of *Slc17a6*, *Kiss1* and *Tac2* in *Slc17a6* KO Kiss1ARH neurons and in control Kiss1ARH cells. The expected base pair (bp) sizes for *Slc17a6*, *Kiss1* and *Tac2* are 194, 120 and 147 bp, respectively. RNA extracted from the medial basal hypothalamic tissue was used as positive (+, with RT) and negative (-, without RT) tissue controls. MM, molecular marker. (**B**) Experimental protocol: AAV1-DIO-ChR2:mCherry was bilaterally injected into ARH of Kiss1Cre:GFP control and *Slc17a6* KO mice, followed by high-frequency photostimulation of Kiss1ARH neurons/terminals and recording of Kiss1ARH neurons. (**C**) high-frequency optogenetic stimulation (20 Hz, 10 s) generated a slow EPSP in an arcuate Kiss1Cre:GFP neuron from OVX, control Kiss1 mice. (**D**), high-frequency response (slow EPSP) in arcuate Kiss1Cre:GFP neurons from OVX, Kiss1Cre:GFP::Slc17a6lox/Δ (KO) mice.(**E**) high-frequency response (slow EPSP) in arcuate Kiss1Cre:GFP neuron from E2-treated, OVX control Kiss1 mice. Inset shows full amplification of sEPSP. (**F**) high-frequency response in arcuate Kiss1Cre:GFP neuron from E2-treated, OVX Kiss1Cre:GFP::Slc17a6lox/Δ mice. Inset shows full amplification of sEPSP. (**G**) summary of the effects of vGluT2 deletion on slow EPSP amplitude: (one-way ANOVA, effect of treatment, $F_{(3, 50)}$=14.13, p<0.0001; Newman-Keuls' Multiple-comparison test, *** or ###, indicates p<0.005). Although knockout of vGluT2 did not significantly diminish the slow EPSP amplitude in OVX females, it did attenuate the response in E2-treated, OVX females (Unpaired t-test, $t_{(25)}$=2.735, p=0.0113). a-a, p<0.05.
DOI: https://doi.org/10.7554/eLife.35656.021

The following source data is available for figure 11:

**Source data 1.** Data for *Figure 11G*.
DOI: https://doi.org/10.7554/eLife.35656.022

treated, OVX control and E2-treated, OVX *Slc17a6* KO mice revealed a significantly reduced slow EPSP in *Slc17a6* KO as compared to control Kiss1ARH neurons (*Figure 11E,F,G*). Therefore even though the slow EPSP is generated mainly by Tac2 in OVX females (*Qiu et al., 2016*), glutamate may contribute to the slow EPSP in Kiss1ARH neurons in E2-treated, OVX females.

## Deletion of *Slc17a6* in Kiss1 neurons abrogates fast glutamatergic responses in Kiss1AVPV/PeN, NPY and POMC neurons

As described above, low-frequency (0.5 Hz) stimulation of Kiss1Cre:ChR2 neurons/fibers activates POMC, NPY/AgRP and Kiss1AVPV/PeN neurons via direct ionotropic glutamatergic input from Kiss1ARH neurons (*Figures 4*, *5*, *6* and *12*). Therefore, we used whole-cell, voltage-clamp recordings in Kiss1AVPV/PeN, POMC and NPY/AgRP neurons from female *Kiss1CreGFP::Slc17a6lox/Δ* mice and found

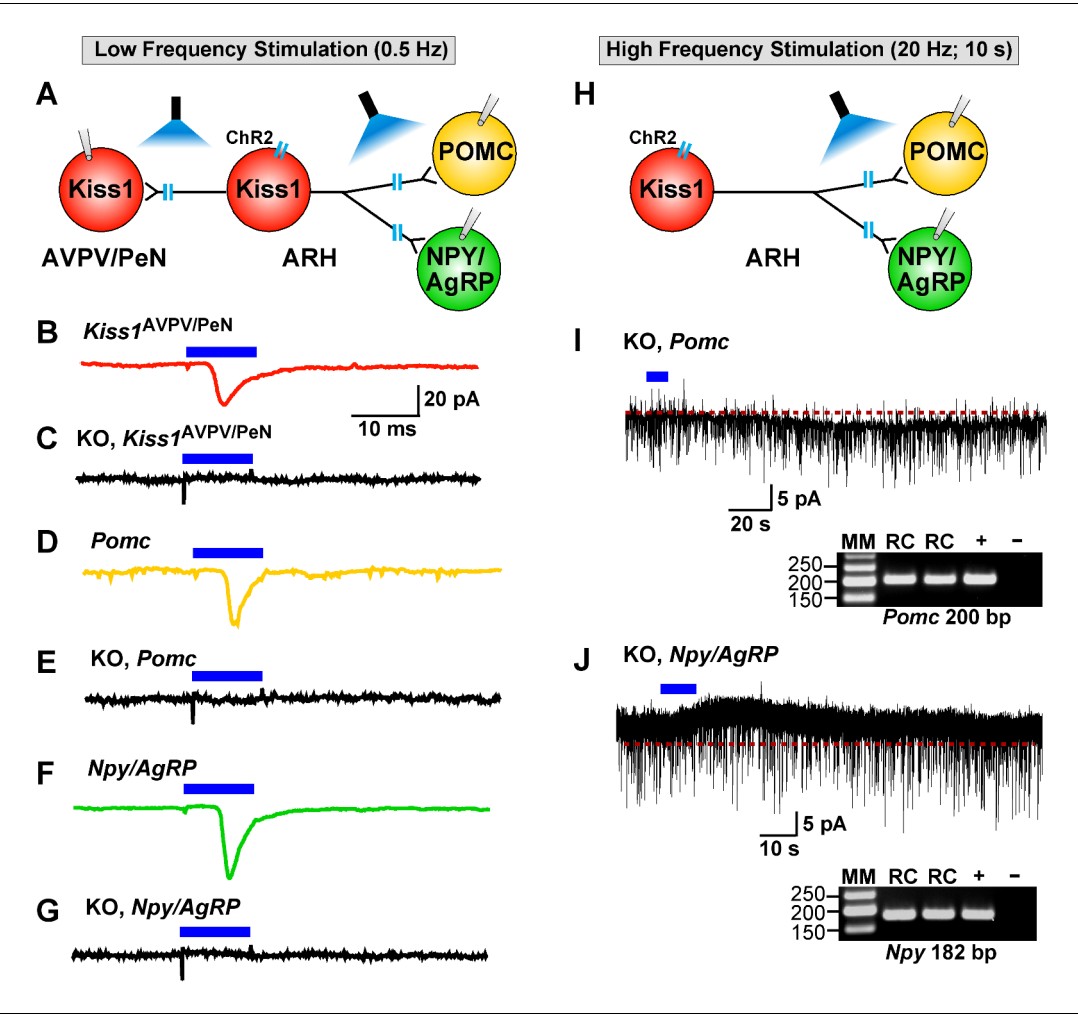

**Figure 12.** Deletion of Slc17a6 in Kiss1[ARH] neurons abrogates fast glutamatergic responses in Kiss1[AVPV/PeN], POMC and NPY/AgRP neurons. (**A**) Experimental protocol: AAV1-DIO-ChR2:mCherry (or YFP) was bilaterally injected into ARH of Kiss1[Cre:GFP] mice. Thereafter, low-frequency photostimulation of the terminals of Kiss1[ARH] neurons were done, and postsynaptic responses in Kiss1[AVPV/PeN], POMC or NPY/AgRP neurons were recorded. (**B, C**) whole-cell, voltage clamp ($V_{hold}$ = −60 mV) recordings in Kiss1[AVPV/PeN] neurons show that low-frequency optogenetic stimulation (0.5 Hz) evoked fast glutamatergic postsynaptic responses in control female Kiss1[Cre] mice (**B**, red trace), but failed in Kiss1[AVPV/PeN] cells (n = 10) from Kiss1[Cre:GFP]::Slc17a6[lox/Δ] mice (**C**, black trace). (**D,E**) and **F,G**) similarly, the response could be induced in POMC neurons (**D**, yellow trace) or NPY/AgRP neurons (**F**, green trace) from control Kiss1[Cre:GFP] mice, but abrogated in POMC neurons (**E**, black trace) (n = 28) or NPY/AgRP neurons (**G**, black trace) (n = 30) from Kiss1[Cre:GFP]::Slc17a6[lox/Δ] mice. (**H**) Experimental protocol: high-frequency photostimulation of the terminals of Kiss1[ARH] neurons and recording of POMC or NPY/AgRP neurons. (**I**) high-frequency stimulation (20 Hz, 10 s) of arcuate Kiss1 neurons from Kiss1[Cre:GFP]::Slc17a6[lox/Δ] mice evoked a small inward current (2.8 ± 0.5 pA, n = 7) in POMC neurons (identified *post hoc* by scRT-PCR, gel inset). (**J**) likewise, high-frequency stimulation evoked a small outward current (4.0 ± 1.6 pA, n = 4) in NPY/AgRP neurons (identified *post hoc* by scRT-PCR, gel inset). Insets show scRT-PCR *post hoc* identification of representative recorded POMC and NPY neurons. RC, recorded cells; +, positive tissue control reacted with RT; -, negative tissue control reacted without RT; MM, molecular marker.

DOI: https://doi.org/10.7554/eLife.35656.023

that low-frequency optogenetic stimulation (0.5 Hz) failed to evoke a fast glutamatergic postsynaptic response in Kiss1[AVPV/PeN] neurons (*Figure 12C*), POMC neurons (*Figure 12E*) or NPY/AgRP neurons (*Figure 12G*). Interestingly, high-frequency stimulation (20 Hz, 10 s) still evoked a small residual inward current (2.8 ± 0.3 pA, n = 7) in POMC neurons (identified *post hoc* by scRT-PCR, inset) from

E2-treated, OVX KO females (*Figure 12H,I* versus 7B,F). Likewise, high-frequency stimulation (20 Hz, 10 s) evoked a small outward current (4.0 ± 1.6 pA, n = 4) in NPY/AgRP neurons (identified *post hoc* by scRT-PCR, inset) from E2-treated, KO females (*Figure 12H,J* versus 7D,F). Although, selective peptidergic inhibitors for potentially blocking kisspeptin-mediated responses in NPY/AgRP and POMC neurons are not available, we suspect, based on previous findings (*Fu and van den Pol, 2010*), that these evoked postsynaptic responses were generated by kisspeptin release following high-frequency stimulation of Kiss1$^{ARH}$ neurons.

To further study the actions of kisspeptin on NPY/AgRP and POMC neurons, we applied kisspeptin directly on these neurons. Kisspeptin (200 nM) inhibited firing and hyperpolarized NPY$^{GFP}$ neurons even in the presence of the GABA$_A$ blocker bicuculline (10 μM) (*Figure 13A,B*). Also, similar to the GABA$_B$ receptor agonist baclofen, kisspeptin induced an outward, albeit smaller current with a reversal potential close to E$_K^+$ (−90 mV) (*Figure 13C*). In addition to binding to the Kiss1 receptor (GPR54), kisspeptin is also known to bind to and activate neuropeptide FF receptors 1 and 2 (NPFFR1 and NPFFR2), and these Gαi/o coupled receptors are both expressed in the ARH (*Elhabazi et al., 2013*; *Rønnekleiv et al., 2014*). We used scRT-PCR to document the expression of *Npffr1* mRNA in NPY neurons (*Figure 13D*). Finally, the RFamide-related peptide-3 (RFRP-3) (10 μM), a selective agonist for NPFFR1 and NPFFR2 (*Bonini et al., 2000*), hyperpolarized and inhibited the firing of NPY/AgRP neurons (*Figure 13E*). The I/V plot of the RFRP-3 induced current showed a reversal potential close to E$_K^+$ (−85 mV) (*Figure 13F*). Therefore, it is probable that kisspeptin and RFRP3 activate the same receptor on NPY/AgRP neurons.

We also explored the actions of kisspeptin on POMC neurons after first identifying that *Kiss1r* mRNA (GPR54) is expressed in a subpopulation of these neurons using scRT-PCR (*Figure 14A*). In agreement with a previous publication (*Fu and van den Pol, 2010*), we found that kisspeptin (200 nM) depolarized and increased the firing frequency of POMC neurons (*Figure 14B*). In addition, the I/V plot showed that kisspeptin activated a non-selective cationic channel that reversed at −10 mV (*Figure 14C*).

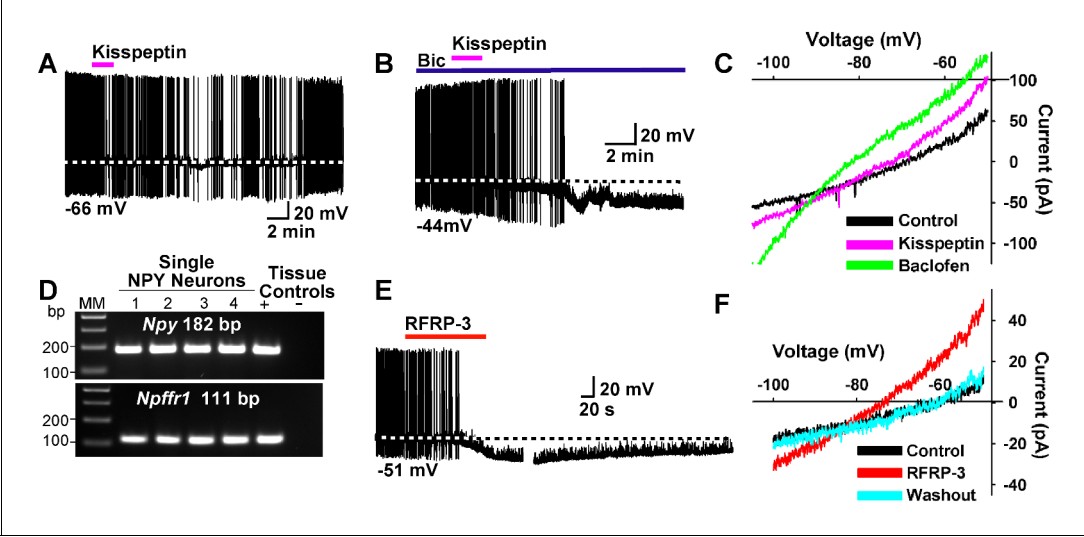

**Figure 13.** Kisspeptin and RFRP-3 inhibit NPY neurons. (**A,B**) kisspeptin (200 nM) inhibited the firing and hyperpolarized NPY neurons even in the presence of GABA$_A$ blocker bicuculline (BIC) (10 μM). (**C**) Similar to GABA$_B$ receptor agonist baclofen (10 μM), kisspeptin induced an outward, albeit smaller, current with a reversal potential at E$_{K+}$ (−90 mV). (**D**) scRT-PCR expression of *Npffr1* in NPY/AgRP neurons. (**E**) RFRP-3 (10 μM), selective agonist for NPFFR1/NPFFR2, hyperpolarized and inhibited firing in NPY neurons. (**F**) the I/V plots of the RFRP-3 current showed a reversal potential close to E$_{K+}$ (−85 mV).
DOI: https://doi.org/10.7554/eLife.35656.024

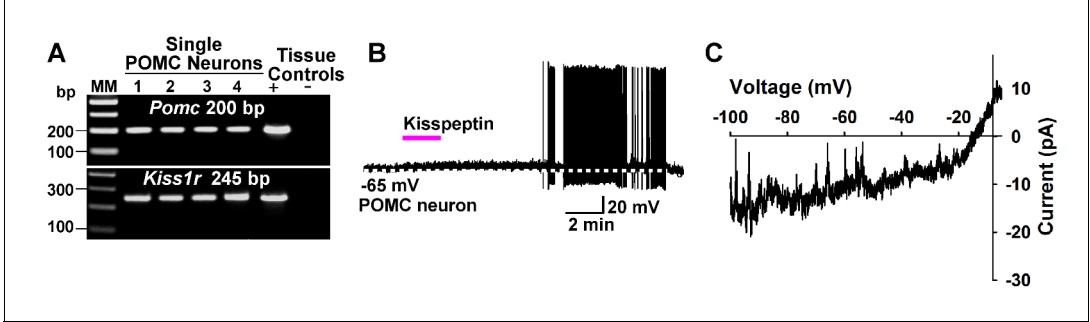

**Figure 14.** Kisspeptin excites POMC neurons by activating a non-selective cation conductance. (**A**) representative gel illustrating the scRT-PCR expression of *Kiss1r* (GPR54) transcript in POMC neurons. (**B**) kisspeptin (200 nM) depolarized and increased firing of POMC neurons (n = 8). (**C**) I/V (digital subtraction of control I/V from I/V with kisspeptin using a Cs$^+$-based internal solution; see Materials and Methods) showed that kisspeptin activated a non-selective cationic channel that reversed at −10 mV.

DOI: https://doi.org/10.7554/eLife.35656.025

## Female mice lacking *Slc17a6* in Kiss1 neurons develop a condition place preference for sucrose

*Ad libitum* access to standard, low-fat mouse chow did not affect body weight in mice lacking *Slc17a6* in Kiss1 neurons versus control females over the limited time-course of our study (data not shown). Therefore, we hypothesized that the lack of glutamate release from Kiss1$^{ARH}$ neurons might increase motivation for palatable food in *Slc17a6* KO females due to diminished regulatory synaptic input onto POMC, NPY/AgRP neurons or other target neurons. For this analysis we used a conditioned place preference (CPP) paradigm (*Figure 15*), that has been used extensively to evaluate

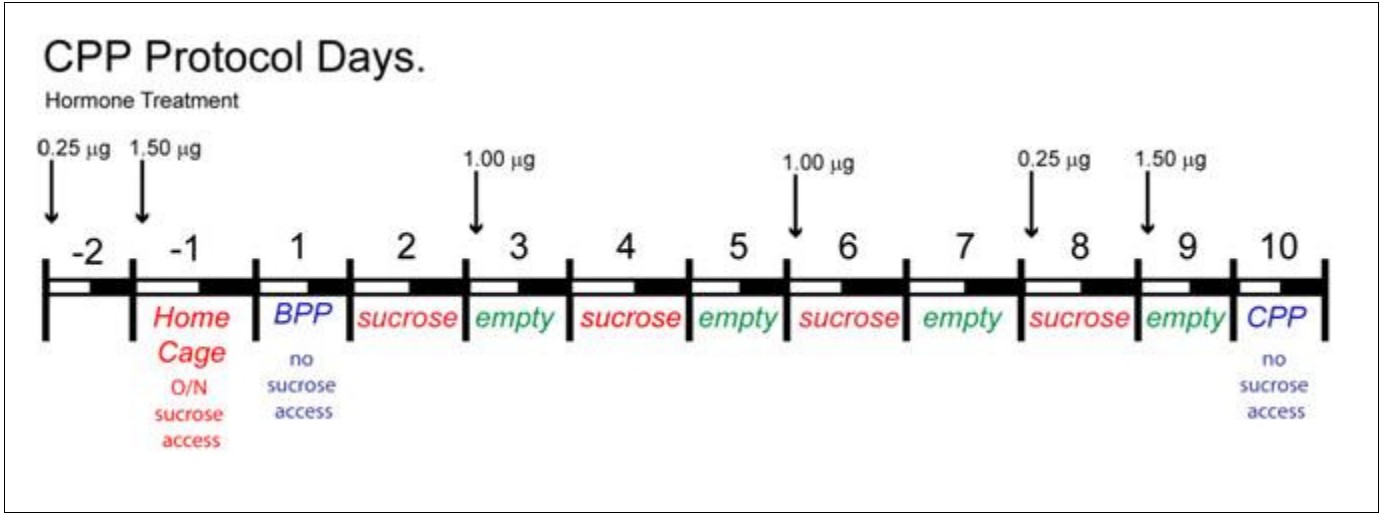

**Figure 15.** Protocol for inducing CPP with sucrose. The protocol for conditioning and preference testing consisted of four phases over the course of 11 days (sucrose habituation, a baseline place preference (BPP) test, sucrose conditioning, and a conditioned place preference (CPP) test). Food-motivated behavior was assessed during the dark cycle using an unbiased procedure. On the day before BPP, Day −1, sucrose habituation occurred where mice received sucrose pellets overnight (O/N) in their home cage to prevent neophobia. The initial BPP (black vs. white chamber) was assessed on Day 1 in a three-chamber place preference apparatus and the chamber pairing was assigned in an unbiased manner. During sucrose conditioning, mice were given access to sucrose-filled (CS+, Days 2, 4, 6, 8) or empty (CS-, Days 3, 5, 7, 9) lids on alternating days. Mice were given access to sucrose-filled lids in one chamber (e.g. white), then on alternating days they were presented with empty lids in the other chamber (e.g. black). Mice were tested for acquisition of a CPP to sucrose on Day 10, which was indicated by increased time spent in the sucrose-conditioned chamber. Animals were fed *ad lib* standard low-fat chow in their home cage throughout the study. For cyclical estradiol treatment, animals were given a priming (0.25 µg) and a surge (1.5 µg) dose of 17β-estradiol Benzoate (E2) at 9 AM prior to the BPP and prior to the CPP as indicated. During the sucrose-conditioning (phase 3), the animals were treated twice with a 1 µg maintenance dose of E2.

DOI: https://doi.org/10.7554/eLife.35656.026

drugs of abuse as well as the preference for palatable food (*Cunningham et al., 2006*; *Harris et al., 2005*; *Prus et al., 2009*; *Sinclair et al., 2017*). While several groups have used sucrose pellets/solutions to produce a CPP (*Alderson et al., 2001*; *Duarte et al., 2003*; *Papp et al., 2002*) many investigators have found food restriction as a necessary adjuvant (*Baunez et al., 2005*; *Figlewicz et al., 2001*). However, even with short-term food deprivation, homeostatic processes can be engaged that prevent selective study of hedonic pathways, as evidenced by the ability of standard chow to produce a CPP following 20 h/day of food restriction (*Papp et al., 2002*; *Popik et al., 2003*). Therefore, to study hedonic feeding, we used *ad-libitum* fed mice and tested them during the dark phase when mice normally eat most of their food.

Food-motivated, behavioral analysis revealed that E2-treated, OVX Kiss1 female mice failed to exhibit a change in place preference for sucrose, a natural food reward (*Figure 16A*, *Figure 16—figure supplement 1*). In contrast, E2-treated, OVX *Slc17a6* KO mice significantly increased their occupancy of the sucrose-paired chamber (*Figure 16A*). E2-treated, OVX Heterozygous ($Slc17a6^{+/lox}::Kiss1^{CreGFP/+}$) mice showed a trend to increase their occupancy in the sucrose-paired chamber (*Figure 16A*). In addition, we measured the amount of sucrose consumed by each experimental group during conditioning. The E2-treated, OVX Kiss1 controls did not increase their sucrose intake at any time (*Figure 16B*, *Figure 16—figure supplement 1*). The *Slc17a6* KO females slightly increased their sucrose intake on day 6 of the test (third day of sucrose exposure; *Figure 15*) and this was significantly augmented by day 8 of the test (fourth day of sucrose exposure) (*Figure 16B*, *Figure 15*). The *Slc17a6* Het mice exhibited a smaller, but significant increase in sucrose consumption on day 6 of the test (*Figure 16B*). Together, these findings indicate that E2-treated, OVX Kiss1 control females had a consistent diminished motivation for sucrose, whereas abolishing glutamate release from Kiss1 neurons enhanced the motivational response for sucrose. Interestingly, attenuating glutamate release from Kiss1 neurons in *Slc17a6* Het females nearly recapitulated the phenotype of the *Slc17a6* KO animals suggesting a gene-dosage/threshold effect. We also measured the body weight gain over the ten-day period of the CPP test. Although both the *Slc17a6* KO and Het mice gained weight in comparison to the E2-treated, OVX Kiss1 controls, the difference was only statistically significant in the *Slc17a6* Het mice (*Figure 16C*). Based on these findings, we measured the mRNA expression of *Slc17a6* in Kiss1[ARH] neurons in the heterozygous E2-treated, OVX females (*Slc17a6 Het*) as compared to control E2-treated, OVX Kiss1[Cre] females. As illustrated (*Figure 16D*), *Slc17a6* mRNA in Het mice was ~35% of that of controls (*Slc17a6* Het mice, 5 pools each from 4 animals; Kiss1 control mice, 5 pools each from 5 animals), and *Slc17a6* mRNA was not detected in Kiss1 neurons from *Slc17a6* KO females. Interestingly intact males, which have reduced Slc17a6 mRNA expression in Kiss1[ARH] neurons and reduced glutamate release compared to castrates (*Nestor et al., 2016*) and E2-treated females, demonstrated a motivation for sucrose based on CPP (*Figure 16—figure supplement 1*).

## Deletion of *Slc17a6* in Kiss1 neurons does not alter the estrous cycle

We have shown previously and currently that Kiss1[ARH] neurons exhibit direct communication with Kiss1[AVPV/PeN] neurons via glutamate, which project to and excite GnRH neurons via kisspeptin release (*Qiu et al., 2016*). Therefore, we hypothesized that glutamate released from Kiss1[ARH] neurons would be involved in the excitation of Kiss1[AVPV/PeN] neurons and the induction of the GnRH surge. As a consequence, deletion of *Slc17a6* in Kiss1[ARH] neurons might lead to disruption of the estrous cycle and the GnRH surge and thus affect fertility. However, we found that the estrous cycle was normal in *Slc17a6 KO* females (one-way ANOVA; p=0.68; comparing cycle length between: wild types (WT), n = 5, 5.55 ± 0.42; Het, n = 3, 4.89 ± 0.48; and KO, n = 7, 5.43 ± 0.44), although the staging of the cycle was not confirmed by steroid hormone measurements. Similarly, the time to conception when placed with a fertile male was not different between the three groups (one-way ANOVA; p=0.68 comparing time (days) to conception between: WT, n = 9, 1.75 ± 0.40; Het, n = 3, 2.83 ± 1.64; KO, n = 8, 1.89 ± 0.67).

## Discussion

It is well known that the peptide neurotransmitters in Kiss1[ARH] neurons are negatively regulated by E2, and these neurons are responsible for pulsatile release of GnRH and reproduction (*Oakley et al., 2009*). Our current findings that E2 increases the expression of *Slc17a6* and

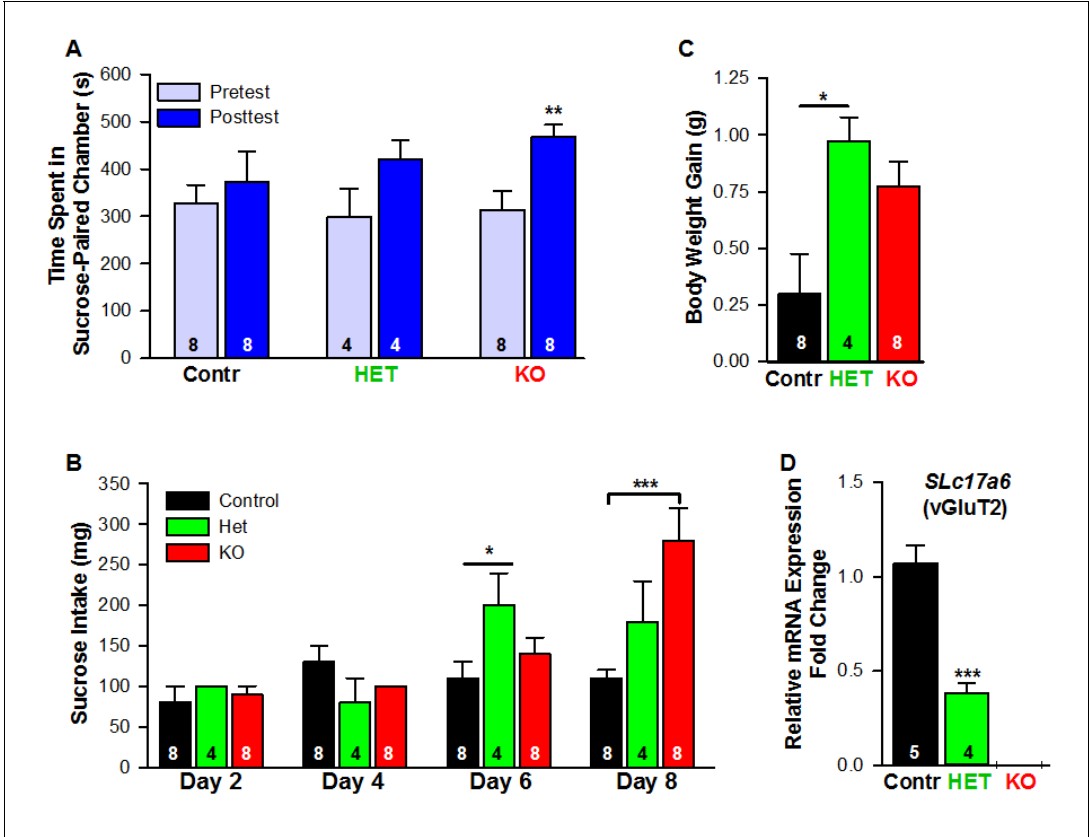

**Figure 16.** Female mice lacking *Slc17a6* in Kiss1^ARH neurons develop a conditioned place preference for sucrose. (**A**) Time spent in sucrose-paired chamber by control Kiss1 female mice (n = 8), *Slc17a6* Het (n = 4) and *Slc17a6* KO Kiss1 females (n = 8) was measured during the Pretest (Day 1, Baseline Place Preference) and the Posttest (Day 10, Conditioned Place Preference). All animals were OVX and E2-treated, and had free access to standard mouse chow in their home cage throughout the study. After sucrose conditioning, *Slc17a6* KO mice developed a preference for the sucrose-paired chamber (Bonferroni *post hoc* test, p=0.001). *Slc17a6* Het mice displayed a trend to develop a preference for the sucrose-paired chamber (Bonferroni *post hoc* test, p=0.086). Control Kiss1^Cre female mice, however, failed to develop a preference (Bonferroni *post hoc* test, p=0.619). [Also, see **Figure 16—figure supplement 1A** for comparison between E2-treated, OVX Kiss1 female and intact Kiss1 male mice]. Two-way ANOVA: main effect of experimental group ($F_{(2,17)}$ = 0.298, p=0.746), main effect of protocol day ($F_{(1,17)}$ = 20.34, p=0.0003), and interaction ($F_{(2,17)}$ = 2.33, p=0.128); **p<0.01.(**B**) Sucrose consumption during the CPP. Sucrose intake (mg) was measured during the four sucrose-paired days (Days 2, 4, 6, and 8). *Slc17a6* KO mice slightly increased their sucrose intake on Day 6 and this was significantly increased by Day 8 (Bonferroni *post hoc* test, p<0.0001). *Slc17a6* Het mice displayed a smaller, but significant increase in sucrose intake on Day 6 (Bonferroni *post hoc* test, p=0.0464). [Also, see **Figure 16—figure supplement 1B** for comparison between E2-treated, OVX Kiss1 females and intact Kiss1 males]. Two-way ANOVA: main effect of experimental group ($F_{(2,17)}$ = 3.788, p=0.0436), main effect of protocol day ($F_{(3,51)}$ = 12.75, p<0.0001), and interaction ($F_{(6,51)}$ = 5.763, p<0.0001). *p<0.05, Het mice versus Kiss1 control; ***p<0.001, *Slc17a6* KO mice versus Kiss1 control. (**C**) Body weight-gain during the ten-day CPP period. Despite that both the *Slc17a6* KO and Het mice gained weight in comparison to control Kiss1 mice, only *Slc17a6* Het mice were significantly different (Bonferroni *post hoc* test, p=0.0312, *Slc17a6* Het vs Kiss1 control; p=0.066, *Slc17a6* KO vs Kiss1 control;). One-way ANOVA: main effect of experimental group ($F_{(2,17)}$ = 5.232, p=0.017). *p<0.05, Het mice versus Kiss1 control. (**D**) Quantitative real time PCR measurement of *Slc17a6* in Kiss1^ARH neuronal pools from control Kiss1^Cre:GFP mice (5 Kiss1 neurons in each pool and 5 pools from each of 5 animals) and *Slc17a6* Het Kiss1 mice (5 Kiss1 neurons in each pool and 5 pools from each of 4 animals). *Slc17a6* KO Kiss1 mice did not express *Slc17a6* in Kiss1^ARH neurons. (Unpaired t-test, $t_{(7)}$ = 5.791, p=0.0007). ***p<0.001, Het mice versus Kiss1 control.

DOI: https://doi.org/10.7554/eLife.35656.027

The following source data and figure supplements are available for figure 16:

**Source data 1.** CPP Time Spent: Ovx Kiss2 Female Mice (n = 8); Ovx HET Female Mice (n = 4); Ovx KO Female Mice (n = 8) for **Figure 16A**.
DOI: https://doi.org/10.7554/eLife.35656.030
**Figure supplement 1.** Studies documenting that intact male mice develop a conditioned place preference for sucrose.
DOI: https://doi.org/10.7554/eLife.35656.028
**Figure supplement 1—source data 1.** Intact Kiss2 Male Mice (n = 15) and Kisspeptin E2 Female Mice (n = 8) for panels A and B.
DOI: https://doi.org/10.7554/eLife.35656.029

glutamate release reveal that the amino acid and peptide neurotransmitters are regulated differentially by E2 in Kiss1[ARH] neurons in females in contrast to our findings in males (*Nestor et al., 2016*). We also found that E2 increased the expression of T-type calcium and h-currents in female Kiss1[ARH] neurons, which led to increased neuronal excitability, concomitant with E2-induced inhibition of the expression of the peptide neurotransmitter mRNAs *Kiss1*, *Tac2* and *Pdyn*. Optogenetic activation of Kiss1[ARH] neurons revealed direct frequency-dependent glutamatergic and peptidergic neurotransmission to POMC and AgRP neurons in females, an indication that Kiss1[ARH] neurons may play a role in regulating feeding behavior. The glutamatergic outputs were lost in females with conditional ablation of *Slc17a6* in Kiss1[ARH] neurons, whereas excitatory and inhibitory kisspeptin responses could be evoked in POMC and NPY/AgRP neurons, respectively. Experiments in vivo revealed that Slc17a6 KO females did not gain weight on normal mouse chow, but the motivation to ingest sucrose was increased in females lacking vGluT2 in Kiss1 neurons. Overall, these and other findings support the idea that Kiss1[ARH] neurons may provide E2- and frequency- dependent signals not only to POMC and NPY/AgRP neurons, but also to Kiss1[AVPV/PeN] neurons to help coordinate feeding and reproduction in females (*Figure 17*).

Kiss1[ARH] neurons are known to co-express the peptide neurotransmitters Kiss1, Tac2 and Dynorphin (*Goodman et al., 2007*; *Lehman et al., 2013*; *Navarro et al., 2009*). Using sc-qPCR we observed that *Tac2*, and not *Kiss1*, is by far the most highly expressed peptide mRNA in Kiss1[ARH] neurons. Even in the presence of high physiological levels of E2, the mRNA expression of *Tac2* was many-fold higher than *Kiss1* under the same treatment. Since Tac2 plays a key role in synchronous firing of Kiss1[ARH] neurons (*Qiu et al., 2016*), which underlies the pulsatility of GnRH release that

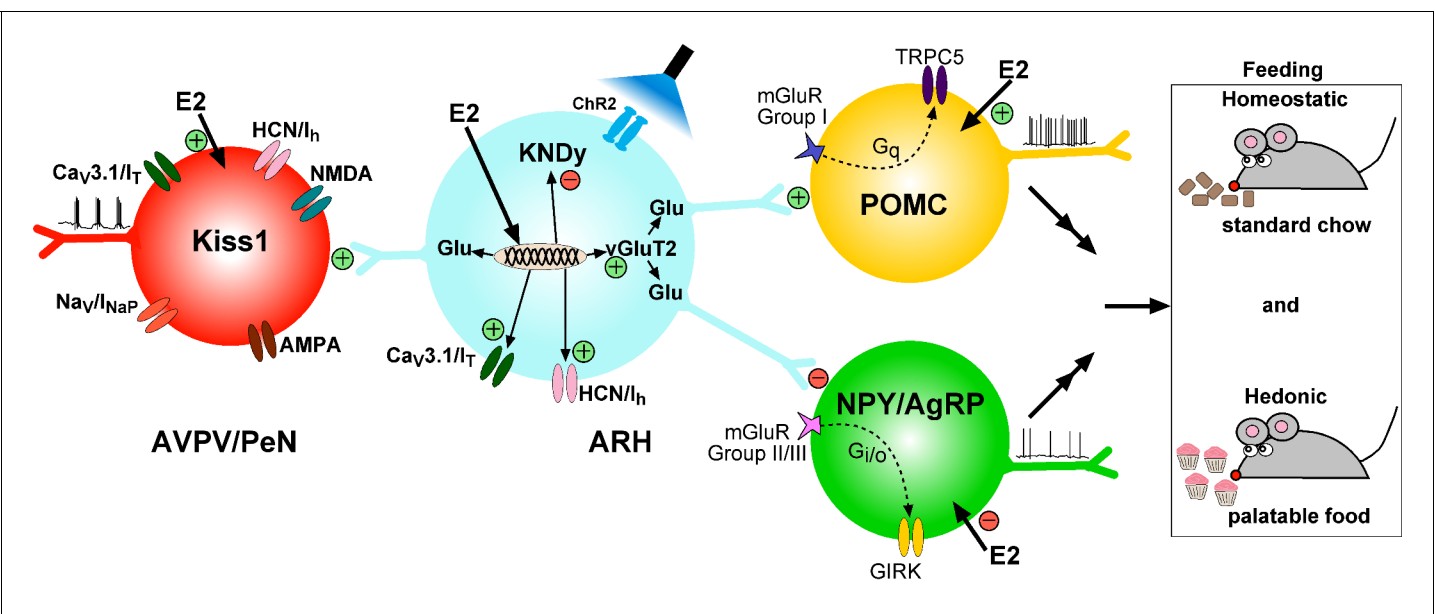

**Figure 17.** Working Model. KNDy (Kisspeptin, NKB, Dynorphin) neurons in the ARH express Ca$_V$3 (I$_T$) and HCN (I$_h$) channels (currents) that are upregulated by E2 and contribute to increased excitability of Kiss1[ARH] neurons. Kiss1[AVPV/PeN] neurons also express Ca$_V$3 (I$_T$), HCN (I$_h$) and Nav (I$_{NaP}$) channels that are highly up-regulated by E2 along with *Kiss1* mRNA expression. Notably, E2 induces spontaneous, repetitive burst firing activity in Kiss1[AVPV/PeN] neurons necessary for the release of GnRH (*Wang et al., 2016*; *Zhang et al., 2015*). E2 also directly excites POMC neurons via inhibition of GIRK current, but inhibit NPY/AgRP neurons via activation of GIRK current (*Kelly and Rønnekleiv, 2015*). These congruent actions of E2 on POMC and NPY/AgRP neurons contribute to the control of homeostatic feeding. High frequency photo-stimulation (focal light stimulation of channel rhodopsin, ChR2) in Kiss1[ARH] neurons releases glutamate to further excite POMC neurons via mGluRs group I and inhibit NPY/AgRP neurons via mGluRs group II/III; and excite Kiss1[AVPV/PeN] neurons via NMDA/AMPA receptors. Ablating *Slc17a6* from Kiss1[ARH] neurons, results in the abrogation of glutamate release onto POA and ARH neurons. The lack of glutamate release from Kiss1[ARH] neurons appears to have little or no effect on estrous cyclicity, an indication that the direct effects of E2 to increase the excitability of Kiss1[AVPV/PeN] neurons is sufficient to drive the reproductive cycle. However, E2-treated *Slc17a6* KO Kiss1 mice develop a condition place preference for sucrose indicative of positive motivational effect of sucrose in these females.

DOI: https://doi.org/10.7554/eLife.35656.031

drives pituitary LH secretion (*Clarkson et al., 2017*), it is not surprising that *Tac2* was found to be the most highly expressed peptide in Kiss1[ARH] neurons.

Given that the peptide neurotransmitters in Kiss1[ARH] neurons are primarily down-regulated by E2, at least in rodents, the Kiss1[ARH] neurons are believed to be under inhibitory control by E2 and are important for negative-feedback regulation of GnRH and LH secretion (*Lehman et al., 2013*; *Navarro et al., 2009*; *Smith et al., 2005*). However, our past (*Gottsch et al., 2011*) and current findings that these Kiss1[ARH] neurons express T-type calcium and pacemaker h-currents and are keenly sensitive to excitation by glutamate are indications that these neurons have pacemaker electrophysiological properties similar to other CNS neurons (*Bal and McCormick, 1993*; *Lüthi and McCormick, 1998*). Additionally, in contrast to the neuropeptides, E2 increased *Slc17a6* mRNA expression in Kiss1[ARH] neurons and increased glutamate release onto Kiss1[AVPV/PeN] neurons in the POA, and onto POMC and NPY/AgRP neurons in the ARH. This is a clear indication that the amino acid and peptide neurotransmitters are regulated differentially by E2 in Kiss1[ARH] neurons in females. Interestingly, we have reported that *Slc17a6* mRNA expression in Kiss1[ARH] neurons and the probability of glutamate release are decreased along with the neuropeptides in intact versus castrated males (*Nestor et al., 2016*). Therefore, there is a significant male/female difference in sex-steroid regulation of glutamate signaling by Kiss1[ARH] neurons (*Nestor et al., 2016*). This could underlie the sex differences in feeding behavior between males and females, namely that testosterone increases feeding in castrated males, whereas E2 reduces feeding in OVX females and during the peri-ovulatory phase of the estrous cycle when E2 levels are maximal (*Asarian and Geary, 2006*; *Asarian and Geary, 2013*).

The mechanism by which E2 signals in Kiss1[ARH] neurons to inhibit the expression of neuropeptides appears to be via ERα, given that ablation of ERα in Kiss1[ARH] neurons prevents E2-induced suppression of *Kiss1* mRNA, and also global KO of ERα prevents suppression of *Tac2* and *Pdyn* by E2 in the ARH (*Dubois et al., 2016*; *Yang et al., 2017*). The mechanism by which E2 enhances glutamatergic expression and transmission in Kiss1[ARH] neurons is currently unknown. Potentially, E2 acting via different E2 receptors and/or signaling pathways in Kiss1[ARH] neurons could be responsible for the diverse actions of E2 (decrease in the expression of neuropeptides versus the increase in expression of excitatory ion channels and glutamate release) similar to what has been reported previously in hippocampal neurons and in hypothalamic NPY/AgRP neurons (*Boulware et al., 2005*; *Smith et al., 2013*, *Smith et al., 2014*).

The excitatory glutamatergic inputs to NPY/AgRP neurons play a key role in the response to fasting, although the origin of this excitatory input has not been determined (*Liu et al., 2012*). Similarly, it has been shown that glutamatergic neurons other than POMC neurons within the arcuate nucleus are responsible for a fast acting satiety signal to suppress feeding even after a 24 hr fast in male mice (*Fenselau et al., 2017*). In addition, our current findings using optogenetic stimulation of Kiss1[ARH] neurons in vitro have documented direct excitatory projections to both POMC and NPY/AgRP neurons by low-frequency stimulation, whereas high-frequency stimulation activated POMC and inhibited NPY/AgRP neurons in both males (*Nestor et al., 2016*) and females (current findings). The low-frequency excitation was blocked by TTX, but reinstated by application of the potassium channel blocker 4-AP, which is biophysical evidence of direct synaptic input from Kiss1[ARH] neurons to POMC and NPY/AGRP neurons in agreement with our previous findings in males (*Nestor et al., 2016*). High-frequency optogenetic stimulation (20 Hz) of Kiss1[ARH] neurons, which mimics a firing rate that is observed in these ARH neurons in vivo (*Moss et al., 1975*), excited POMC neurons. This appears to be generated by an activation of group I metabotropic glutamate receptors 1 and 5, both of which we have documented are expressed in POMC neurons. Also, we found that the group I mGluR agonist DHPG excited POMC neurons in the presence of TTX and ionotropic glutamate receptor blockers. Collectively these data demonstrate that there is a direct excitatory input to POMC neurons from Kiss1[ARH] neurons via glutamate actions on ionotropic and metabotropic receptors. In contrast, we found that high-frequency stimulation generated an inhibition of NPY/AgRP neurons, which was mediated by glutamatergic type II/III metabotropic receptors. Indeed, we found that NPY/AgRP neurons express *Grm7* mRNA (encoding mGluR7), and to a lesser extent *Grm2* mRNA (encoding mGluR2) and that the high frequency response was inhibited by the mGluR7 antagonist ADX71743. In addition, these neurons were inhibited by bath application of the group II mGluR agonist DCG-IV and the group III mGluR agonist AMN082 in the presence of synaptic blockade by TTX, and ionotropic glutamate and GABA$_A$ inhibitors, providing evidence for a direct inhibitory Kiss1[ARH]

glutamatergic input to NPY/AgRP neurons. The response in POMC neurons to group I metabotropic agonist DHPG, which activates a TRPC current (*Tozzi et al., 2003*), was not different between oil- and E2-treated, OVX females. In contrast, the GIRK current in NPY/AgRP neurons, which was activated by the group II and the group III metabotropic glutamate receptor agonists, was augmented by E2-treatment, as was the mRNA expression of mGluR7. Congruent with these findings, we have shown that E2-treatment increases KCNQ 5 channel expression and the corresponding inhibitory M-current in NPY/AgRP neurons (*Roepke et al., 2011*) and augments postsynaptic $GABA_B$ receptor coupling ( *Smith et al., 2013*, *Smith et al., 2014*). Clearly, E2 inhibits the orexigenic NPY/AgRP neurons via multiple mechanisms and thereby helps to prevent hyperphagia (*Roepke et al., 2010*).

The evidence that Kiss1[ARH] neurons are involved in the regulation of GnRH and LH pulsatility via their peptide neurotransmitters is well documented (*Clarkson et al., 2017*; *Goodman et al., 2013*; *Navarro et al., 2009*; *Qiu et al., 2016*; *Wakabayashi et al., 2010*). Although it is known that *Slc17a6* is expressed in Kiss1[ARH] neurons (*Cravo et al., 2011*; *Nestor et al., 2016*), the role of glutamate neurotransmission from these neurons had not been elucidated. Currently, we deleted vGluT2 (*Slc17a6*) in Kiss1 neurons and found that glutamate release was completely abrogated in the ARH population of Kiss1 neurons. Given that we have documented previously that Kiss1[AVPV/PeN] neurons do not express Slc17a6 (*Qiu et al., 2016*) and do not release glutamate, we believe that the KO would be specific for Kiss1[ARH] neurons. However, Kiss1 neurons are expressed in other brain regions, including the medial amygdala and BNST (*Lehman et al., 2013*), and although *Slc17a6* has not yet been documented in these neurons, we cannot rule out the possibility that vGluT2 has been deleted in extra-hypothalamic kisspeptin neurons.

Within the Kiss1[ARH] neurocircuitry, the lack of glutamate resulted in a diminished slow EPSP in E2-treated animals. The slow EPSP is attenuated by E2 treatment (*Qiu et al., 2016*), and it was further reduced in E2-treated *Slc17a6* KO females. Given that the slow EPSP underlies Tac2-dependent engagement of synchronous activity in Kiss1[ARH] neurons (*Qiu et al., 2016*), these findings support the idea that glutamate may play a role in Kiss1[ARH] synchronous firing activity and LH pulsatility in the presence of high circulating levels of E2, when peptide neurotransmitters are low and glutamate levels are high in female Kiss1[ARH] neurons. Interestingly, females with ablation of vGluT2 in Kiss1 neurons appeared to exhibit a normal ovulatory cycle, an indication that glutamatergic neurotransmission from Kiss1 neurons may not be necessary to support reproductive function. However, deletion of ERα in all kisspeptin neurons (KERKO) significantly attenuates glutamatergic synaptic input (spontaneous EPSCs) to the Kiss1[AVPV/PeN] neurons, which could be due to lack of E2-stimulated glutamate release from Kiss1[ARH] neurons (*Wang et al., 2017*). The net result is an increase in LH pulse frequency in KERKO mice, which was proposed to represent a lack of negative feedback (*Wang et al., 2017*). Clearly further experiments, including deletion of *Slc17a6* specifically in adult females, which would prevent potential developmental compensation, are necessary to determine whether glutamate and/or peptide neurotransmission from Kiss1[ARH] neurons to Kiss1[AVPV/PeN] neurons are involved in reproductive functions.

High-frequency stimulation of Kiss1[ARH] neurons in slices from E2-treated OVX *Slc17a6* KO females induced a low-amplitude, inward current in POMC neurons and a low-amplitude, outward current in NPY/AgRP neurons, an indication of residual excitatory and inhibitory inputs, respectively to these neurons. Given that the high frequency stimulation has been documented to induce peptide release (*Qiu et al., 2016*), this is an indication that kisspeptin release from Kiss1[ARH] neurons evoked the excitation of POMC neurons and inhibition of NPY/AgRP neurons. In agreement with previous studies (*Fu and van den Pol, 2010*), bath application of kisspeptin activated POMC neurons most likely by signaling via the Kiss1 receptor, GPR54, which is expressed in POMC neurons. Similarly, kisspeptin inhibited firing and hyperpolarized NPY/AgRP neurons via activation of a GIRK current. The Kiss1 receptor is not expressed in NPY/AgRP neurons. However, NPFFR1 is expressed, and its agonist RFRP-3 inhibited firing and hyperpolarized NPY/AgRP neurons via activation of a GIRK current. Based on evidence that both RFRP-3 and kisspeptin bind to and activate NPFFR1 (*Bonini et al., 2000*), we propose that the two different neuropeptides inhibit NPY/AgRP neurons by action on the same receptor. Collectively, these data indicate that Kiss1[ARH] neurons may use both kisspeptin and glutamate to activate POMC neurons but inhibit NPY/AgRP neurons, and compensatory kisspeptin input could be one of the reasons why deleting vGluT2 was not effective to alter body weights on a normal chow diet over the time-course of our study.

As the Kiss1 *Slc17a6* KO females did not appear to gain weight on standard low-fat mouse chow diet, we used a place preference paradigm to determine the motivation for sucrose in *Slc17a6* KO as compared to control Kiss1 and *Slc17a6* Het females (*Prus et al., 2009*). These experiments revealed a preference for the sucrose-paired chamber by the E2-treated, OVX mice lacking *Slc17a6* in Kiss1 neurons compared to Kiss1[Cre] and Het females. The *Slc17a6* Het females did not exhibit a CPP to sucrose, but did consume more sucrose in comparison to control Kiss1 females. Importantly, the E2-treated *Slc17a6* Het females expressed low levels of *Slc17a6* mRNA in Kiss1[ARH] neurons, an indication that glutamate release was attenuated (*Herman et al., 2014*), which could be why the *Slc17a6* Het females behaved more like *Slc17a6* KO than control Kiss1 animals thereby exhibiting a behavioral gene-dosage/threshold effect. While reward-seeking behaviors have not been specifically associated with Kiss1[ARH] neurons, NPY/AgRP and POMC neurons are involved in hedonic feeding in addition to controlling homeostatic food intake (*Betley et al., 2013*; *Hayward et al., 2006*; *Lippert et al., 2014*; *Pandit et al., 2014*,*Pandit et al., 2015*; *Rubinstein and Low, 2017*; *Stuber and Wise, 2016*) Although it is clear that homeostatic feeding is regulated by NPY/AgRP and POMC neurons, the motivational drive to eat palatable foods are more complex behaviors that are regulated by multiple interconnected neural systems (*Chen et al., 2016*; *Denis et al., 2015*; *Fenselau et al., 2017*; *Rossi and Stuber, 2018*). Therefore, further analyses of the neural circuits underlying these physiological responses are essential for understanding such complex behavior. Regardless, our findings have revealed a novel role for glutamate neurotransmission by Kiss1[ARH] neurons (*Figure 17*), which appears to be sex-specific based on our findings that glutamate signaling is regulated differently in female and male (*Nestor et al., 2016*) mice.

# Materials and methods

### Key resources table

| Reagent type (species)or resource | Designation | Source or reference | Identifiers | Additional information |
|---|---|---|---|---|
| Strain, strain background (*M.Musculus*) | C57BL/6J | The Jackson laboratory | RRID :IMSR_JAX:000664 | |
| Genetic reagent (*M.Musculus*) | Kiss1[Cre:GFP] | Dr. Robert A Steiner; University of Washington; PMID:21933870 | RRID:IMSR_JAX:017701 | Full nomenclature: Kiss1[tm1.1(cre/EGFP)Stei] |
| Genetic reagent (*M.Musculus*) | Kiss1[Cre:GFP] version 2 (V2) | Dr. Richard D. Palmiter; University of Washington; PMID: 29336844 | | |
| Genetic reagent (*M.Musculus*) | Pomc[EGFP] | Dr. Malcolm J. Low; University of Michigan; PMID:11373681 | RRID:IMSR_JAX:009593 | Full nomenclature: Tg(Pomc-EGFP) 1Low |
| Genetic reagent (*M.Musculus*) | Npy[GFP] | Dr. Brad Lowell; Harvard University; PMID:19357287 | JAX stock #006417 | Full nomenclature: Tg(Npy-hrGFP) 1Lowl |
| Genetic reagent (*Adeno-associated virus*) | AAV1-Ef1α-DIO-ChR2:YFP | Dr. Stephanie L. Padilla; University of Washington; PMID: 25429312 | | |
| Genetic reagent (*Adeno-associated virus*) | AAV1-Ef1α-DIO-ChR2: mCherry | Dr. Stephanie L. Padilla; University of Washington; PMID: 25429312 | | |
| Antibody | Anti-mCherry (rabbit polyclonal) | Abcam | Abcam: ab167453 RRID:AB_2571870 | (1:10,000) |

*Continued on next page*

Continued

| Reagent type (species)or resource | Designation | Source or reference | Identifiers | Additional information |
|---|---|---|---|---|
| Antibody | Anti-kisspeptin (rabbit polyclonal) | Dr. Alain Caraty Universite Francois -Rabelais Tours; PMID:16621281 | No. 564 AB_2622231 | (1:2500) |
| Antibody | Goat anti-rabbit conjugated to Alexa 594 | Life Technologies (ThermoFisher) | Cat. No.: # A-11037 RRID: AB_2534095 | (1:500) |
| Antibody | Goat anti-rabbit conjugated to Alexa 488 | Life Technologies (ThermoFisher) | Cat. No.: # A-11034 RRID: AB_2576217 | (1:500) |

## Mice

All procedures conducted with animals were according to the National Institutes of Health Guide for the Care and Use of Laboratory Animals with approval for all of the animal use procedures from the Oregon Health and Science University (OHSU) and/or the University of Washington (UW) Animal Care and Use Committees.

Kiss1$^{Cre:GFP}$ (**Gottsch et al., 2011**), Kiss1$^{Cre:GFP}$ version 2 (V2) (**Padilla et al., 2018**), Pomc$^{EGFP}$ (**Cowley et al., 2001**), Npy$^{GFP}$ (**van den Pol et al., 2009**), Kiss1$^{Cre:GFP}$::Pomc$^{EGFP}$ and Kiss1$^{Cre:GFP}$:: Npy$^{GFP}$ female mice were housed under constant temperature (21–23°C) and 12 hr light, 12 hr dark cycle schedule (lights on at 0600 and lights off at 1800 hr), with free access to food (Lab Diets 5L0D) and water. Where specified, Kiss1$^{Cre:GFP}$, Kiss1$^{Cre:GFP}$::Pomc$^{EGFP}$ and Kiss1$^{Cre:GFP}$::Npy$^{GFP}$ mice received viral injections to express channelrhodopsin 2 (ChR2) in Kiss1$^{ARH}$ neurons (see below). Kiss1$^{Cre:GFP}$::Pomc$^{EGFP}$ and Kiss1$^{Cre:GFP}$::Npy$^{GFP}$ mice were produced by crossing heterozygous Kiss1$^{Cre:GFP}$ mice with Pomc$^{EGFP}$ and Npy$^{GFP}$ mice, respectively.

## Kiss1$^{Cre:GFP}$ version 1 (V1) and Kiss1$^{Cre:GFP}$ V2

The offspring from Kiss1$^{Cre;GFP}$ V1 animals crossed with a reporter line like lacZ, sometimes showed ectopic expression (**Gottsch et al., 2011**). Subsequently, the Palmiter group produced Kiss1$^{Cre:GFP}$ V2 animals in which the Cre was less efficient (**Padilla et al., 2018**), thereby significantly decreasing the likelihood of ectopic expression (**Song and Palmiter, 2018**). However, Kiss1$^{Cre:GFP}$ V2 animals express less GFP and, therefore, the individual Kiss1 neurons are more difficult to see in a slice preparation unless injected with Cre-dependent ChR2-mCherry (or ChR2-YFP) to label Kiss1 neurons. Currently, we have used both Kiss1$^{Cre:GFP}$ lines and we have found no differences between the two in terms of the molecular, electrophysiological and optogenetic characterization.

## AAV delivery

Viruses were prepared at the University of Washington according to published methods (**Gore et al., 2013**). Fourteen to twenty-one days prior to each experiment, Kiss1$^{Cre:GFP}$, Kiss1$^{Cre:GFP}$, V2, Kiss1$^{Cre:GFP}$::Pomc$^{EGFP}$ or Kiss1$^{Cre:GFP}$::Npy$^{GFP}$ female mice (>60 days old) received bilateral ARH injections of a Cre-dependent adeno-associated viral (AAV; serotype 1) vector encoding ChR2 fused to yellow fluorescent protein, YFP (AAV-EF1α-ChR2:YFP) or mCherry (AAV-EF1α-ChR2:mCh) as described (**Nestor et al., 2016**). Briefly, using aseptic techniques, anesthetized mice (1.5% isoflurane/$O_2$) were placed in a Kopf stereotaxic apparatus (Tujunga, CA) and received a medial skin incision to expose the surface of the skull. Two holes were drilled into the skull at designated coordinates from bregma (x: ± 0.30 mm; y: -1.200 mm). For the viral injections, a glass micropipette (Drummond Scientific #3-000-203-G/X; Broomall, PA) was fabricated with a Narishige PE-2 puller and beveled (tip diameter = 45 μm), filled with mineral oil and loaded with an aliquot of AAV using a Nanoject II (Drummond Scientific). The pipette tip was positioned at x: -0.30 mm; y: -1.200 mm and lowered to z: -5.800 mm (surface of brain z = 0.0 mm). The AAV (2.0 × 10$^{12}$ particles/ml) was injected at a rate of 100 nl/min (300 nl total), raised to −5.700 mm for a second injection (200 nl total) and then left in place for 10 min post-injection before the pipette was slowly removed from the brain. The other hemisphere injection was done at the same y and z coordinates, but at x: +0.30 mm. The skin incision was closed using skin adhesive, and each mouse received analgesia

(Carprofen; 5 mg/kg; subcutaneous) for two days post-operation. These arcuate injections only labeled Kiss1 cell bodies in the ARH as evaluated using immunocytochemistry and scRT-PCR (see *Figure 6*) (*Nestor et al., 2016*) (*Qiu et al., 2016*).

## Ovariectomy (OVX) and estradiol treatment

We were interested in exploring the actions of preovulatory levels of 17β-Estradiol (E2) on neuro-transmitter expression, neuronal activity and behavior in females. Therefore, we used an E2-treatment regimen that we have documented induces a preovulatory surge of luteinizing hormone (LH) in GnRH[GFP] and Kiss1[GFP] females (*Bosch et al., 2013*; *Zhang et al., 2013*). When necessary, at least seven days prior to each experiment, ovaries were removed as described previously while under iso-flurane inhalation anesthesia (Piramal Enterprises Limited, Andhra Pradesh, India) (*Zhang et al., 2013*). Each mouse received analgesia (Carprofen; 5 mg/kg; subcutaneous) on the day of operation. 17β-Estradiol benzoate (E2) treatments were as described previously (*Zhang et al., 2013*). Briefly, each animal was injected on day 5 following OVX with 0.25 μg E2, followed on day 6 with 1.50 μg E2 and used for experiments on day 7. High circulating (proestrous) levels of 17β-estradiol (E2) were verified by the uterine weights (>100 mg) at the time of the hypothalamic slice preparation.

## qPCR and scRT-PCR

qPCR and scRT-PCR were conducted as previously described (*Bosch et al., 2013*). Briefly, the ARH was microdissected from 240 μm basal hypothalamic coronal slices (3–4 slices/per mouse) from OVX oil- and estradiol-treated female *Kiss1*[Cre:GFP] mice (n = 5–7 animals/group; qPCR), POMC[EGFP], and NPY/AgRP[GFP] mice (n = 5 animals each/group; qPCR). The dispersed cells were visualized using a Leitz inverted fluorescent microscope, patched, and then harvested with gentle suction to the pipette using a Xenoworks digital micromanipulator system (Sutter Instrument; Novato, CA) and expelled into a siliconized 0.65 ml microcentrifuge tube containing a solution of 1X Invitrogen Super-script III Buffer, 15 U of RNasin (Promega), 10 mM dithiothreitol and diethylpyrocarbonate-treated water in a total of 5 μl for a single cell (1 cell/tube for scRT-PCR) or 8 μl for pooled cells (5 cells/tube for qPCR). cDNA synthesis was performed on single cells and pools of 5 cells as previously described (*Bosch et al., 2013*) and stored at −20°C. Controls included non-fluorescent cells, artificial CSF (aCSF), single cells without reverse transcriptase (RT) and tissue controls with and without RT. Pri-mers for the genes that encode for Kiss1 (*Kiss1*), NKB (*Tac2*), NKB receptor (*Tacr3*) Dynorphin (*Pdyn*), POMC (*Pomc*), NPY (*Npy*), AgRP (*Agrp*), vGluT2 (*Slc17a6*), vGAT (*Slc32a1*), mGlur1 (*Grm1*), mGlur2 (*Grm2*) mGlur3 (*Grm3*), mGlur5 (*Grm5*), mGlur7 (*Grm7*), GAPDH (*Gapdh*), β-actin (*Actb*), $Ca_V3.1$ (*Cacna1g*), HCN1 (*Hcn1*) and HCN2 (*Hcn2*) were designed using Clone Manager software (Sci Ed Software) to cross at least one intron-exon boundary and optimized as previously described (*Bosch et al., 2013*; *Nestor et al., 2016*). See *Table 1* for primer sequences and optimal annealing temperature for all genes, as well as amplification efficiency for each gene used for quantitative anal-ysis. Note that for qPCR, the annealing temperature for all genes was set at 60°C. Primers for qPCR were further tested for efficiency ($E = 10^{(-1/m)} - 1$; *Table 1*) (*Livak and Schmittgen, 2001*; *Pfaffl, 2001*). qPCR was performed on a Quantstudio 7 Flex Real-Time PCR System (Life Technolo-gies) using the Power Sybrgreen (Life Technologies) mastermix method according to established protocols (*Bosch et al., 2013*). The comparative ΔΔCT method (*Livak and Schmittgen, 2001*; *Pfaffl, 2001*) was used to determine values from duplicate or triplicate samples of 4 μl for the target genes and 2 μl for the reference gene. The relative linear quantity was determined using the $2^{-\Delta\Delta CT}$ equation (*Bosch et al., 2013*). In order to determine the relative expression levels of target genes in Kiss1[ARH] neurons obtained from OVX and estradiol-treated animals, the mean ΔCT for the target genes from the OVX female samples were used as the calibrator, and the data were expressed as *n*-fold change in gene expression normalized to the reference gene *Gapdh* (Kiss1 neurons) or *Actb* (NPY neurons) relative to the calibrator. For quantification differences between the mRNA expres-sion of *Tac2* (NKB), *Pdyn*, *Kiss1* and *Slc17a6* in Kiss1[ARH] neurons, the mean Δ CT for the target gene *Tacr3* from the OVX female samples were used as the calibrator. For quantification of *Grm7* in NPY/AgRP neurons, the mean ΔCT for the target genes from the OVX female samples were used as the calibrator as described above. scRT-PCR was performed on 3 μl of cDNA in a 30 μl reaction volume and amplified 50 cycles using a C1000 Thermal Cycler (Bio-Rad).The PCR product was visualized with ethidium bromide on a 2% agarose gel.

## Electrophysiology and optogenetics

Coronal brain slices containing the ARH from AAV1-EF1α-DIO-ChR2:YFP injected $Kiss1^{Cre:GFP}$ female mice, $Pomc^{EGFP}$ female mice, $Npy^{GFP}$ female mice, AAV1-EF1α-DIO-ChR2:mCh injected $Kiss1^{Cre:GFP}::Pomc^{EGFP}$ female mice and AAV1-EF1α-DIO-ChR2:mCh injected $Kiss1^{Cre:GFP}::Npy^{GFP}$ female mice were prepared as previously described (*Nestor et al., 2016*; *Qiu et al., 2003*). Whole-cell patch recordings were performed in voltage-clamp and current-clamp using an Olympus BX51W1 upright microscope equipped with video-enhanced, infrared-differential interference contrast (IR-DIC) and an Exfo X-Cite 120 Series fluorescence light source. Electrodes were fabricated from boro-silicate glass (1.5 mm outer diameter; World Precision Instruments, Sarasota, FL) and filled with a normal internal solution (in mm): 128 potassium gluconate, 10 NaCl, 1 $MgCl_2$, 11 EGTA, 10 HEPES, 2 ATP, and 0.25 GTP (pH was adjusted to 7.3–7.4 with 1N KOH, 290–300 mOsm). Pipette resistances ranged from 3 to 5 MΩ. In whole cell configuration, access resistance was less than 20 MΩ; access resistance was 80% compensated. To display reversal potential and rectification characteristics of the ligand-activated currents, I–V plots were constructed by voltage ramps from −100 mV to 0 mV applied over 2 s from a holding potential of − 60 mV. For some experiments measuring the ramp current-voltage (I-V), $K^+$-gluconate in the normal internal solution was replaced with $Cs^+$-gluconate (pH 7.35 with CsOH) (*Qiu et al., 2010*), and the extracellular solution contained $Na^+$, $K^+$, $I_h$ (HCN), $Ca^{2+}$, and $GABA_A$ channel blockers (in mM: NaCl, 126; 4-aminopyridine, 5; KCl, 2.5; $MgCl_2$, 1.2; CsCl, 2; $CaCl_2$, 1.4; $CoCl_2$, 1; nifedipine, 0.01; HEPES, 20; NaOH, 8; glucose, 10; tetrodotoxin (TTX), 0.001; picrotoxin, 0.1). For optogenetic stimulation, a light-induced response was evoked using a light-emitting diode (LED) 470 nm blue light source controlled by a variable 2A driver (ThorLabs, Newton, NJ) at 0.5–20 Hz with the light path directly delivered through an Olympus 40 × water immersion lens. For high-frequency (20 Hz) stimulation the length of stimulation was 10 s (*Qiu et al., 2016*). Electrophysiological signals were digitized with Digidata 1322A (Molecular Devices, Foster City, CA), and the data were analyzed using p-Clamp software (version 9.2, Molecular Devices). The liquid junction potential was corrected for all data analyses. After recording, the cytosol of non-fluo-rescent recorded cells was harvested and used for *post-hoc* identification by scRT-PCR using the same protocol as for the dispersed single cells (see above).

## Deletion of Slc17a6 in Kiss1$^{Cre:GFP}$ neurons

The deletion of *Slc17a6* specifically in Kiss1 neurons was accomplished according to previously described procedures (*Hnasko et al., 2010*) by breeding $Slc17a6^{lox/lox}$ mice with $Slc17a6^{+/\Delta}::Kiss1\text{-}Cre:GFP/+$ mice. $Slc17a6^{\Delta/lox}::Kiss1^{Cre:GFP/+}$ (KO) and $Slc17a6^{+/lox}::Kiss1^{Cre:GFP/+}$ (Het mice) were deter-mined using the following primers: CGC AGC CAT TCA CCT GTC TAA G; AAA GGT CCT GGA TCA GAG CAG G; and CAG TGT GCT GTA ACT GAG ATA GT (*Song and Palmiter, 2018*). Approx-imately 25% of the offspring from this cross were conditional knockout $Slc17a6^{\Delta/lox}::Kiss1^{Cre:GFP/+}$ and ~ 25% were heterozygous (Het) $Slc17a6^{+/lox}::Kiss1^{Cre:GFP/+}$. The Cre-negative mice would be $Slc17a6^{+/lox}$ (WT) or $Slc17a6^{\Delta/+}$ (Het). PCR of tail DNA that detected both the lox and the Δ (null) alleles of *Slc17a6* was used to detect unexpected (ectopic) recombination that sometimes occurs with this cross (*Song and Palmiter, 2018*). Mice with ectopic recombination were not used for experiments.

## Estrous cycle and fertility

Prior to monitoring the estrous cycle, females were habituated to handling for at least 3 days, and during the study, each cage was *only* handled by the investigator. To track the estrous cycle, a vagi-nal lavage with sterile 0.1M PBS was performed daily between 0700 hr to 0800 hr, and evaluated by light microscopy for the appearance of: cornified epithelial cells, leukocytes, and nucleated epithelial cells. The loss of leukocytes and presence of an abundance of nucleated cells was scored as proes-trus, while the transition to primarily cornified epithelial cells was scored as estrus. Note one acyclic mouse was excluded from both the wild type and homozygous groups. For fertility measurements, females were mated with a sexually-experienced wild type male on day 0. Litters were tracked from day 16 through 30. The length of time until conception was calculated as the number of days until parturition minus a 20 day gestation period. Note, two wild type and two homozygous animals were excluded because they did not give birth within the 30 day trial.

# Conditioned place preference (CPP) behavioral assay for vGluT2 deleted and control Kiss1 animals

*Slc17a6* Het (*Slc17a6$^{+/lox}$::Kiss1$^{Cre:GFP/+}$*, n = 4) and KO (*Slc17a6$^{\Delta/lox}$::Kiss1$^{Cre:GFP/+}$*, n = 8) females were initially injected with AAV1-DIO-ChR2:YFP in the arcuate nucleus as described above in order to do optogenetic experiments in vitro following in vivo conditioned place preference (CPP) tests. Control Kiss1 (Kiss1$^{Cre:GFP/+}$) females (n = 8), destined for CPP analysis, received sham surgery, but without viral injection. In addition, we prepared another set of control females (*Kiss$^{Cre:GFP/+}$*, n = 7), which similar to *Slc17a6* Het and KO females received viral injection prior to the CPP test. Both Kiss1 control groups exhibited similar results in the CPP test, indicating that the presence of unstimulated ChR2::YFP fusion protein had no effect on behavior.

The animals were OVX at the time of viral injections or sham surgery and received equal cyclical (~every 3–5 days) priming 0.25 µg and/or a surge 1.0–1.5 µg dose of 17β-estradiol benzoate (E2) (*Bosch et al., 2013*) prior to and throughout the CPP procedure (see *Figure 15*). The E2-treatment started on days 5–6 following OVX with a 2 µg dose, followed 6–7 days later with another 2 µg dose during the recovery period from viral or sham injections. Thereafter, the cyclical E2-treatment was initiated with a 0.25 µg dose followed the day after with 1.5 µg E2, a treatment shown to induce a LH surge in GnRH mice (*Bosch et al., 2013*) and Kiss1 mice (*Zhang et al., 2013*). As indicated in *Figure 15*, we used this E2-treatment immediately prior to the baseline place preference (BPP; initiated 18–20 days after OVX) and prior to the CPP in all groups. The animals were handled daily over a 7 day period prior to the CPP procedure in order to adapt to the investigator and minimize handling stress. The protocol for inducing CPP with sucrose consisted of four phases over the course of 11 days (sucrose habituation, a BPP test, sucrose conditioning, and a CPP test) as depicted in *Figure 15*. During sucrose habituation (Day −1), mice received sucrose in their home cage to prevent neophobia. Animals received 0.5 g of sucrose reward pellets (TestDiet, Richmond, IN) in a scintillation vial lid left in cages overnight. The next morning cages were changed so no sucrose remained. All behavior testing was conducted in a separate room kept on the same light schedule as the colony room. Mice were transferred into the animal testing room 1 hr prior to dark onset (1700 hr). The CPP behavioral assay was conducted in the dark cycle (beginning at 1800 hr). During the baseline place preference test (Day 1), mice were placed in the central chamber of a three-chamber place preference apparatus (Product Number: MED-CPP-3013, Med Associates Inc, Fairfax, VT) to determine pretest values. Doors were left open for mice to explore all three chambers for 15 min. An unbiased assignment was used where mice for each experimental group Kiss1$^{Cre:GFP}$ controls, vGluT2 KO (*Slc17a6$^{\Delta/lox}$::Kiss1$^{Cre:GFP/+}$*), and heterozygous vGluT2 animals (Slc17a6$^{+/lox}$::Kiss1$^{Cre:GFP/+}$) were subdivided into counterbalanced subgroups based on their assigned conditioning chamber (white or black chamber) (*Cunningham et al., 2003*, *2006*). During the sucrose conditioning phase, mice received sucrose-filled lids (CS+, Days 2, 4, 6, 8) in one chamber or empty lids (CS-, Days 3, 5, 7, 9) in the other chamber on alternating days (*Figure 15*). Mice were confined to a chamber (black or white) with access to sucrose (0.5 g) or without it for 30 min. Lids were stabilized in the chamber with heavy-duty magnets. We had separate sets of lids and magnets for paired (sucrose access) and unpaired days (no sucrose access). To test for acquisition of a CPP for sucrose (Day 10), mice were placed in the central chamber of the apparatus with door open for 15 min. Time spent in the sucrose-paired chamber was recorded pretest (Day 1) and posttest (Day 10). After each animal, the chambers were wiped down with 70% ethanol and paper towels located at the bottom of cages were replaced. At the end of the testing period, animals were returned to the colony room. All animals had free access to normal chow in their home cage.

## Drugs

All drugs were purchased from Tocris Bioscience (Minneapolis, MN) unless otherwise specified, and made up as stock solutions as follows: DL-amino-5-phosphonovaleric acid (AP5) (50 mM), 6-cyano-7-nitroquinoxaline-2,3-dione (CNQX) (10 mM), 4-Aminopyridine (4-AP) (100 mM), 3,5-dihydroxyphenylglycine (DHPG) (50 mM), (2*S*,2′*R*,3′*R*)−2-(2′,3′-Dicarboxycyclopropyl)glycine (DCG-IV) (10 mM), and RFRP 3 (human, 10 mM) were dissolved in H$_2$O. Tetrodotoxin (TTX) was purchased from Alomone Labs (Jerusalem, Israel) (1 mM) and was dissolved in H$_2$O. Bicuculline methiodide (20 mM) was purchased from Sigma-Aldrich (St Louis, MO) and was dissolved in dimethylsulfoxide (DMSO). Picrotoxin (100 mM) was dissolved in DMSO. Kisspeptin-10 [Mouse Kiss1 (110–119)-NH2; Kp-10, from

Phoenix Pharmaceuticals (Belmont, CA)] (100 μM) was dissolved in $H_2O$. The mGluR7 agonist AMN082 dihydrochloride (10 mM) was purchased from Abcam (Cambridge, MA) and was dissolved in DMSO. The mGluR7 antagonist ADX71743 (10 mM) was dissolved in DMSO. Baclofen was purchased from Sigma-Aldrich (St Louis, MO) and dissolved in 0.1 N HCl to a stock concentration of 40 mM. Aliquots of the stock solutions were stored at −20°C until needed.

## Immunocytochemistry

Female Kiss1$^{CreGFP}$ V2 mice, with injection of ChR2-mCherry in the ARH, were prepared for immunocytochemistry as described previously (*Roepke et al., 2011*). Briefly, coronal hypothalamic blocks (2 mm each) were fixed by immersion in 4% paraformaldehyde, cryoprotected in 20% sucrose solution, frozen at −55°C, sectioned coronally on a cryostat at 20 μm, and thaw-mounted on Superfrost Plus slides (Thermo Fisher Scientific). Some sections were rinsed in PB (0.1M phosphate buffer, pH 7.4; all rinses were in PB for at least 30 min), and then incubated for 48 hr at 4° C in rabbit polyclonal antiserum against mCherry (1:10,000; ab167453 Abcam Inc, Cambridge MA). Some preoptic sections were reacted with rabbit polyclonal antiserum against kisspeptin as described previously (*Bosch et al., 2012*) using the Caraty kisspeptin 10 antibody (no. 564; 1:2500) (*Franceschini et al., 2006*). After rinsing, sections stained for mCherry were incubated in goat-antirabbit IgG antibody conjugated to Alexa 594 (1:500; Jackson Immunoresearch). Sections stained for kisspeptin, were incubated in goat-antirabbit IgG antibody conjugated to Alexa 488 (1:500; Jackson Immunoresearch). Following a final rinse, slides were coverslipped with gelvatol containing the anti-fading agent 1,4-diazabicyclo(2,2)octane (Sigma-Aldrich, St Louis, MO).

## Imaging

Photomicrographs of ARH YFP or mCherry expression were acquired using an Olympus BX51W1 upright microscope equipped with a Rolera XR Fast 1394 camera and a Nikon E800 fluorescent microscope equipped with DS-U2 camera. Confocal photomicrographs were acquired using a Zeiss LSM 510 and a Zeiss LSM 780 confocal microscopes, each equipped with 20x and 40x (NA 0.8) APO objectives with Zen software.

## Data Analysis

For qPCR three to seven Kiss1 or NPY/AgRP neuronal pools (5 cells/pool) from each animal were run in duplicate or triplicate for mRNAs that encode for Kiss1, Tac2, Pdyn, Tacr3, vGluT2, vGAT, Ca$_V$3.1, HCN1, HCN2, mGluR7 *and* GAPDH or β-actin and the mean value of each gene from each animal was used for statistical analysis. Data are expressed as Mean ± SEM and were analyzed using a one-way ANOVA and Tukey's multiple comparison *post-hoc* test, or an unpaired student's t-test. Since mRNA for vGAT was below the level of detectability (CT 36) in ARH Kiss1 neurons, we report it as not detectable. For scRT-PCR the number of Kiss1$^{GFP}$, POMC$^{EGFP}$ and NPY/AgRP$^{GFP}$ neurons expressing each transcript was counted for each animal and the mean number of neurons/animal was determined and used for further analysis of Mean, SEM, and percentage expression. Electrophysiology data are expressed as Mean ± SEM and were analyzed using either an unpaired Student's t-test (paired-pulse experiment) or a one-way ANOVA and Newman-Keuls multiple comparison *post-hoc* test (POMC and AgRP pharmacology experiments).

Behavioral data are depicted as Mean ± SEM. CPP and sucrose consumption were analyzed using a two-way ANOVA (experimental group as the between-subject factor and CPP protocol days, pretest and posttest, as the within-subject factor) followed by Bonferroni multiple comparison *post-hoc* test. Body weight gain over the 10 day CPP period was analyzed using a one-way ANOVA followed by Tukey's multiple comparison *post-hoc* test.

## Acknowledgements

We thank Ms. Jessica G Bradner and Ms. Ashley Connors (OHSU) and Ms. Megan Chiang (UW) for excellent help with maintaining the mice breeding colonies. Also, we thank Dr. Christopher Cunningham for advice with designing conditioned place preference (CPP) experiments and analyzing CPP data. This research was funded by National Institute of Health (NIH) grants: R01-NS043330 (OKR), R01-NS038809 (MJK) and R01-DK068098 (OKR and MJK). Confocal microscopy was supported by the P30 NS061800 (PI, S Aicher) grant.

# Additional information

## Competing interests

Richard D Palmiter: Reviewing editor, *eLife*. The other authors declare that no competing interests exist.

## Funding

| Funder | Grant reference number | Author |
|---|---|---|
| National Institutes of Health | R01-DK068098 | Martin J Kelly<br>Oline K Rønnekleiv |
| National Institutes of Health | R01-NS043330 | Oline K Rønnekleiv |
| National Institutes of Health | R01-NS038809 | Martin J Kelly |
| National Institutes of Health | R01-DA024908 | Richard D Palmiter |

The funders had no role in study design, data collection and interpretation, or the decision to submit the work for publication.

## Author contributions

Jian Qiu, Heidi M Rivera, Martha A Bosch, Stephanie L Padilla, Todd L Stincic, Data curation, Methodology, Writing—review and editing; Richard D Palmiter, Resources, Data curation, Funding acquisition, Methodology, Project administration, Writing—review and editing; Martin J Kelly, Oline K Rønnekleiv, Conceptualization, Resources, Data curation, Supervision, Funding acquisition, Validation, Methodology, Writing—original draft, Project administration, Writing—review and editing

## Author ORCIDs

Jian Qiu (iD) https://orcid.org/0000-0002-4988-8587

Richard D Palmiter (iD) https://orcid.org/0000-0001-6587-0582

Martin J Kelly (iD) http://orcid.org/0000-0002-8633-2510

Oline K Rønnekleiv (iD) http://orcid.org/0000-0003-1841-4386

## Ethics

Animal experimentation: Animal experimentation: This study was performed in strict accordance with the recommendations from the National Institutes of Health Guide for the care and use of Laboratory Animals. All animal procedures were conducted according to the approved institutional animal care and use committee (IACUC) protocols (#IP00000585; #IP00000382) at Oregon health and Science University and (#2183-02) at University of Washington. All surgeries were performed using aseptic techniques under isoflurane anesthesia, and every effort was made to minimize suffering.

## Decision letter and Author response

Decision letter https://doi.org/10.7554/eLife.35656.035
Author response https://doi.org/10.7554/eLife.35656.036

# Additional files

## Supplementary files

• Transparent reporting form
DOI: https://doi.org/10.7554/eLife.35656.033

## Data availability

All data generated or analysed during this study are included in the manuscript and supporting files.

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
