## [Decision Letter]

Thank you for submitting your article "Glutamate output from arcuate nucleus Kisspeptin neurons regulates motivational drive for palatable food in females" for consideration by *eLife*. Your article has been reviewed by a Senior Editor, a Reviewing Editor, and three reviewers.. The reviewers have opted to remain anonymous.

The reviewers have discussed the reviews with one another and the Reviewing Editor has drafted this decision to help you prepare a revised submission.

The reviewers agree that your manuscript successfully uses state of the art in vitro methods to extend findings from in vivo experiments to examine the functional relationships between the estrogen-sensitive kisspeptin neurons of the arcuate nucleus and hypothalamic circuits. However, there are several issues that need to be addressed according to the three reviewers:

Essential revisions

1) The major concern is that the conditioned place preference (CPP) conclusions are not supported by the results and should not be overstated in the title. Specifically, the fact that WT OVX-E2 mice do not develop CPP for sucrose raises the issue of whether this test can adequately assess food preference/reward in this model system. Although the authors acknowledge the challenges to measure CPP for sucrose in mice, it is premature to conclude that KO mice have a behavioral phenotype compared to controls not showing the expected response. Alternative CPP protocols (food restriction, high fat-high sucrose pellets) should be considered as well as operant conditioning to measure food reward. Also, simple tests such as a fast-refeeding challenge, food preference or high-fat feeding may help highlight a feeding or metabolic phenotype. Based on the data showing normal energy balance in KO mice, the authors cannot conclude: "Collectively, these data indicate that Kiss1^ARH^ neurons may use both kisspeptin and glutamate to activate POMC but inhibit NPY neurons and thus, participate to maintain normal energy balance".

Accordingly, we suggest that you drop the behavioral work (conditional place preference (CCP) paradigm) and concentrate on addressing the neurobiological issues of specificity of expression etc. leaving behavioral work to a later paper.

2) The physiological relevance of the work is limited because the experiments were done in ovariectomized (OVX) mice in the absence or presence of estradiol (E2) supplementation. Unless the generalization of the work was in the context of abrupt onset of menopause or ovariectomy in patients, it does not seem widely applicable. The study would have had much more breadth if it had been conducted in female mice at key stages of the estrous cycle such as diestrus and proestrus. Typically, studies that intend to use the OVX and the OVX + E2 mouse model to recapitulate the changes in circulating E2 levels occurring during the actual estrous cycle usually implant a silastic capsule filled with E2 in OVX mice to avoid full deprivation in E2 (a condition that never happens in mice) and mimic the diestrus stage.

3) The authors have to be cautious when they state that because *Slc17a6* is not expressed in the kisspeptin neurons of the preoptic region (AVPV/PeN), the lox/Cre mediated ablation of *Slc17a6* using the kisspeptin promoter is specific for kisspeptin neuron in the arcuate nucleus (ARH). there is increasing evidence that the kisspeptin gene is expressed in extra-hyphothalamic areas, some of which could well be involved in controlling motivated behavior. The only way the authors could convincingly show that ARH kisspeptin neurons indeed regulate motivational drive for palatable food, as claimed in the title of the paper, would be to perform experiments in mice in which they selectively manipulate opto/chemo-genetically ARH kisspeptin neurons in mice in vivo (using the tools the authors master well) and hopefully validate their interesting findings in the animal model in which they knockout selectively *Slc17a6* KO only in kisspeptin neurons. In addition, being able to manipulate (stimulate) the activity of ARH kisspeptin neurons in vivo, this would also extend the significance of the present study by determining whether the kisspeptin neurons could actually be the glutamatergic neurons of the ARH that have recently been described to mediate satiety in mice fed at libitum (see Resch et al., 2017, which has not been discussed in the paper).

4) Finally, the authors conclude (Abstract) that the E2-driven increase in excitability and glutamate release from ARH kisspeptin neurons plays a significant role in dampening the motivational drive for palatable food during critical reproductive states. Unfortunately, the authors only performed the experiments in OVX + E2 animals. Because the authors have studied the effect of *Slc17a6* KO in kisspeptin neurons on the estrous cycle and showed that it had no effect on it, they should study condition place preference in these animals in diestrus and proestrus to convincingly corroborate their statement.

Specific comments

1) Results section: The methods for these experiments are not very clear. How did they measure ion channel currents? What channel blockers did the authors use and in what concentration? Stock concentrations are given in the methods but not the final working concentration. For rebound firing – what is the current injection protocol, how long after the hyperpolarizing current do the authors consider a spike to be a rebound spike?

2) Results section: It would be good to see some discussion about why there is a difference in the response of NPY neurons to low frequency vs high frequency stimulation of Kiss1^ARH^ neurons. The authors suggest that the high frequency stimulation is more physiological, in which case I don't understand the point of doing the low frequency stimulation. Do low frequency stimulations not activate the inhibitory metabotropic glutamate receptors? Would there be any physiological conditions where the release of glutamate from Kiss1^ARH^ neurons switches from being inhibitory for NPY to being excitatory?

3) A description of the validation of the injection sites of AVV is not clearly stated.

4) AVPV Kiss1 neurons has been described to project to the ARC (Yeo SH et al., 2016). Because the mouse model expresses Cre in both neuronal populations, even proper injection of AAVs into the ARC could infect AVPV Kiss1 neurons. Therefore, while the lack of action in the presence of TTX indicates that there is synaptic activation, this does not exclude that other AVPV Kiss1 neurons may contact and activate each other when photo-stimulated. Evidence that cell bodies of AVPV Kiss1 neurons (that should be GFP labeled) are devoid of mCherry is necessary.

5) Indicate at what time after OVX or E2 replacement the experiments were performed as well as dose of E2 used.

6) Subsection “High frequency stimulation of glutamate release from Kiss1^ARH^ neurons excites POMC neurons and inhibits NPY neurons via metabotropic receptors”, "revealed a significantly reduced slow EPSP" However, Figure 11G does not show this different to be significant.

7) Has DHPC been used on NPY neurons and DCG-IV on POMC neurons?

8) The authors addressed the role of mGluR2 in NPY neurons, which is expressed only in 10% of them. Therefore, the study of mGluR7, which is more abundant, is missing.

9) E2 dramatically decreases Kiss1 expression (Figure 2) to suggest a decreased Kiss release. As such, it is expected that the Kisspeptin tone onto POMC and NPY neurons would be minimal at high Hz stimulation in OVX-E2 vs. OVX VGluT2 KO mice. Did the authors do these measurements? This should be at least discussed along with potential studies assessing the impact of E2 on responses to Kisspeptin in POMC or NPY neurons.

10) Did the authors test the effect of high Hz stimulation of ARC Kiss1 on AVPV Kiss1 activity in VGluT2 KO mice? This aspect should be at least discussed.

[Editors' note: further revisions were requested prior to acceptance, as described below.]

Thank you for resubmitting your work entitled "Estrogenic-dependent Glutamatergic Neurotransmission from Kisspeptin Neurons Governs Feeding Circuits in Females" for further consideration at *eLife*. Your revised article has been reviewed by a Senior Editor, a Reviewing Editor, and three reviewers.

The manuscript has been largely improved but there is one remaining issue that needs to be addressed before acceptance: could you please add a sentence justifying the use of the OVX and OVX+E2 animal models in the text?

---

## [Author Response]

*The reviewers agree that your manuscript successfully uses state of the art in vitro*in vitro *methods to extend findings from in vivo*in vivo experiments to examine the functional relationships between the estrogen- sensitive kisspeptin neurons of the arcuate nucleus and hypothalamic circuits. However, there are several issues that need to be addressed according to the three reviewers:1) The major concern is that the conditioned place preference (CPP) conclusions are not supported by the results and should not be overstated in the title. Specifically, the fact that WT OVX-E2 mice do not develop CPP for sucrose raises the issue of whether this test can adequately assess food preference/reward in this model system. Although the authors acknowledge the challenges to measure CPP for sucrose in mice, it is premature to conclude that KO mice have a behavioral phenotype compared to controls not showing the expected response. Alternative CPP protocols (food restriction, high fat-high sucrose pellets) should be considered as well as operant conditioning to measure food reward. Also, simple tests such as a fast-refeeding challenge, food preference or high-fat feeding may help highlight a feeding or metabolic phenotype. Based on the data showing normal energy balance in KO mice, the authors cannot conclude: "Collectively, these data indicate that Kiss1^ARH^ neurons may use both kisspeptin and glutamate to activate POMC but inhibit NPY neurons and thus, participate to maintain normal energy balance".Accordingly, we suggest that you drop the behavioral work (conditional place preference (CCP) paradigm) and concentrate on addressing the neurobiological issues of specificity of expression etc. leaving behavioral work to a later paper.

As suggested by the reviewers, we have clarified and added new experiments concerning specificity issues raised in the critique. We have also tempered and modified our CPP conclusions (e.g., see Discussion section). We agree that it would be advantageous to test for body weight gain differences between WT and KO mice fed energy dense food. These experiments are planned with vGluT2 deletion specifically in Kiss1^ARH^ neurons in adult mice using available techniques and will be done in the future in order to more completely characterize the role of Kiss1 neurons in feeding behavior and energy balance. Still, we feel that the current CPP data adds functional significance to the cellular findings, and we would prefer to keep it; but we have deemphasized the data because more research is needed as stated above.

However, based on studies conducted in several mouse models (e.g., Kiss1^Cre^ females, n=8; POMC-EGFP females, n=10; and Kiss1^Cre^ males, n=15), E2-treated OVX female mice do not appear to develop a CPP for sucrose pellets, unless vGluT2 is deleted in Kiss1 neurons. In contrast, intact Kiss1^Cre^ males, which express lower levels of vGluT2 in Kiss1^ARH^ neurons versus E2-treated females, do develop a CPP for sucrose. Data from Kiss1^Cre^ females and males have been added as Figure 15—figure supplement 1. With the E2-treatment paradigm, which is designed to maximize *Slc17a6* mRNA expression differences in Kiss1^ARH^ neurons between the WT and Het as compared to the KO females, we would expect to find no or low preference for sucrose and no body weight gain in E2-treated WT animals. Importantly, WT, Het and KO mice were all OVX and treated equally with E2 and tested simultaneously for sucrose-reinforced CPP according to the unbiased method recommended by Dr. Chris Cunningham, who is a recognized expert on the use of CPP for measuring motivated behavior (Cunningham et al., 2006). The E2-treatment and experimental design have been described in more detail in the Materials and Methods section.

2) The physiological relevance of the work is limited because the experiments were done in ovariectomized (OVX) mice in the absence or presence of estradiol (E2) supplementation. Unless the generalization of the work was in the context of abrupt onset of menopause or ovariectomy in patients, it does not seem widely applicable. The study would have had much more breadth if it had been conducted in female mice at key stages of the estrous cycle such as diestrus and proestrus. Typically, studies that intend to use the OVX and the OVX + E2 mouse model to recapitulate the changes in circulating E2 levels occurring during the actual estrous cycle usually implant a silastic capsule filled with E2 in OVX mice to avoid full deprivation in E2 (a condition that never happens in mice) and mimic the diestrus stage.

We are interested specifically in the differential estradiol regulation of neuropeptides and glutamate neurotransmission in Kiss1^ARH^ neurons as well as the corresponding behavioral effects, which is possible because of novel techniques such as single cell RT-PCR, neuropharmacology/electrophysiology and optogenetics in mouse models. Eventually, we want to explore which estrogen receptors are involved. It would be very difficult, if not impossible, to do these type of experiments in cycling female mice (see response to comment #4). This is the reason why the majority of published behavioral data (including behavioral data in the Lowell lab, 2016, referred to below) are based on experiments done in male mice. Also, it should be noted that during proestrus in females, estrogen, progesterone and inhibin are all circulating at high levels and found to act on the hypothalamus. Therefore, the findings would not be specific for estradiol’s actions.

Concerning the estradiol-treatment paradigm, we started using estradiol implants, which increased Cav3.1, Cav3.3 mRNA expression in GnRH neurons (Zhang et al., 2009). However, with E2 implants we were not able to induce a LH surge in our mouse models. Therefore, we developed an E2-injection procedure utilizing a priming dose followed by a surge-producing dose of E2, which also increases Cav3.1 and 3.3 in GnRH neurons, and with this treatment we have been able to induce a LH surge in GnRH mice (Bosch et al., 2013) and in Kiss1 mice (Zhang et al., 2013). Therefore, we believe that our model is of physiological relevance. We are aware that different steroid replacement mouse models are being used, and sometimes conflicting data are reported (see Liu et al., 2017). Unfortunately, the mouse is not a good model for studying reproduction, and eventually studies have to be done in species with long ovulatory cycle with true follicular and luteal phases such as the guinea pig and non-human primates.

*3) The authors have to be cautious when they state that because Slc17a6 is not expressed in the kisspeptin neurons of the preoptic region (AVPV/PeN), the lox/cre mediated ablation of Slc17a6 using the kisspeptin promoter is specific for kisspeptin neuron in the arcuate nucleus (ARH). there is increasing evidence that the kisspeptin gene is expressed in extra-hyphothalamic areas, some of which could well be involved in controlling motivated behavior. The only way the authors could convincingly show that ARH kisspeptin neurons indeed regulate motivational drive for palatable food, as claimed in the title of the paper, would be to perform experiments in mice in which they selectively manipulate opto/chemo-genetically ARH kisspeptin neurons in mice* in vivo *(using the tools the authors master well) and hopefully validate their interesting findings in the animal model in which they knockout selectively Slc17a6 KO only in kisspeptin neurons. In addition, being able to manipulate (stimulate) the activity of ARH kisspeptin neurons* in vivo*, this would also extend the significance of the present study by determining whether the kisspeptin neurons could actually be the glutamatergic neurons of the ARH that have recently been described to mediate satiety in mice fed at libitum (see Resch et al., 2017, which has not been discussed in the paper).*

While technically true, functionally the deletion of vGluT2 is likely to mainly affect Kiss1^ARH^ neurons. We reported in Qiu et al., 2016 that we did not detect *Slc17a6* mRNA in Kiss1^AVPV/PeN^ neurons using scRT-PCR. Also, using our scRT-PCR procedure, we detect vGAT mRNA in close to 100% of AVPV/PeN Kiss1 neurons. We have not detected glutamate release from these rostral Kiss1 neurons. High frequency stimulation releases kisspeptin (Qiu et al., 2016). Although, Kiss1 neurons have been detected in the amygdala and in the Bed Nucleus of the Stria Terminalis, these Kiss1 neurons have not been reported to express vGluT2 (See Discussion section). However, we have been more cautious and now refer to vGluT2 KO in Kiss1 neurons (rather than Kiss1^ARH^) throughout the manuscript. Also, with this in mind, we have changed the title to the following: “Estrogenic-dependent Glutamatergic Neurotransmission from Kisspeptin Neurons Governs Feeding Circuits in Females”. The Fenselau et al., paper (Lowell lab), which is quite relevant for the current findings, has been added to the Discussion section.

4) Finally, the authors conclude (Abstract) that the E2-driven increase in excitability and glutamate release from ARH kisspeptin neurons plays a significant role in dampening the motivational drive for palatable food during critical reproductive states. Unfortunately, the authors only performed the experiments in OVX + E2 animals. Because the authors have studied the effect of Slc17a6 KO in kisspeptin neurons on the estrous cycle and showed that it had no effect on it, they should study condition place preference in these animals in diestrus and proestrus to convincingly corroborate their statement.

Doing CPP conditioning in intact cycling female mice comparing diestrus versus proestrus animals in control and KO would be difficult and costly. In particular, it has been quite difficult to get the necessary number of *Slc17a6* KO animals of either sex. We would need to have a large group of females in order to have 8-12 diestrus and proestrus females (based on power analysis) from each group to test together for BPP and CPP (on the same day and time). Therefore, as described above, we have developed OVX vehicle- and E2-treated animals as physiological models for E2-induction of the LH surge, and for preventing weight gain in aging females (also see Asarian and Geary, 2013). However, the statement referred to in the Abstract has been modified.

Specific comments:1) Results section. The methods for these experiments are not very clear. How did they measure ion channel currents? What channel blockers did the authors use and in what concentration? Stock concentrations are given in the methods but not the final working concentration. For rebound firing – what is the current injection protocol, how long after the hyperpolarizing current do the authors consider a spike to be a rebound spike?

A citation to a previous paper was inadvertently omitted for the rebound spike measurements and is now included in the legend to Figure 2 (Zhang et al., 2013). We have added the drug concentrations for CNQX (10 μM) and AP5 (50 μM) to the legend for Figure 7, and the baclofen concentration (10 μM) for Figure 13. For Figure 14, we have included a reference (Qiu et al., 2010) in the Materials and methods section for the internal Cs^+^ concentration.

2) Results section: It would be good to see some discussion about why there is a difference in the response of NPY neurons to low frequency vs high frequency stimulation of Kiss1^ARH^ neurons. The authors suggest that the high frequency stimulation is more physiological, in which case I don't understand the point of doing the low frequency stimulation. Do low frequency stimulations not activate the inhibitory metabotropic glutamate receptors? Would there be any physiological conditions where the release of glutamate from Kiss1^ARH^ neurons switches from being inhibitory for NPY to being excitatory?

The low frequency stimulation is used for verifying a direct synaptic connection between Kiss1^ARH^ and NPY/AgRP neurons. The high frequency stimulation probably reflects the physiological activity based on in vivo recordings (e.g., Moss et al., 1975). It is thought that mGluRs are extra-synaptic such that with the high frequency generated glutamate release there is “spillover” from the synaptic cleft to activate these receptors (Watanabe et al., 2003; Nietz et al., 2017.). This has been discussed in the Results section. The physiological conditions when Kiss1^ARH^ neuronal glutamate release would switch from being inhibitory to excitatory for NPY neurons has not yet been studied specifically. However, during high E2 conditions, such as on proestrus, animals are getting ready to mate and reproduce and at this stage of the cycle it has been shown in many animal models including primates that feeding is suppressed (see Asarian and Geary, 2006, 2013). A potential mechanism is Kiss1^ARH^ glutamatergic inhibitory output to NPY neurons. On the other hand, if animals are food deprived, it has been shown that NPY neurons receive excitatory ionotropic glutamate input (Liu et al., 2012), some of which could come from Kiss1^ARH^ neurons. This has now been added to the Discussion section.

3) A description of the validation of the injection sites of AVV is not clearly stated.

In the Materials and methods section, we have described the validation of the injection site. We have published previously in males injected with AAV1-DIO-ChR2-YFP in the ARH, that 100% of the YFP neurons expressed Kiss1 mRNA (Nestor et al., 2016). In females, we have published that arcuate AAV1-DIO-ChR2-mCherry-labeled Kiss1 cells are co-expressed with Kiss1^GFP^ cells (Qiu et al., 2016). We refer to these publications in the Materials and methods section. In addition, we have done a thorough neuroanatomical analysis comparing neuronal projections following ARH and AVPV/PeN viral injections in our Kiss1 animal model. These anatomical data are being prepared for publication elsewhere. Also see response to specific comment # 4.

4) AVPV Kiss1 neurons has been described to project to the ARC (Yeo SH et al., 2016). Because the mouse model expresses Cre in both neuronal populations, even proper injection of AAVs into the ARC could infect AVPV Kiss1 neurons. Therefore, while the lack of action in the presence of TTX indicates that there is synaptic activation, this does not exclude that other AVPV Kiss1 neurons may contact and activate each other when photo-stimulated. Evidence that cell bodies of AVPV Kiss1 neurons (that should be GFP labeled) are devoid of mCherry is necessary.

See response to specific comment #3. Also, we have published that photostimulation of Kiss1^AVPV/PeN^ neurons caused auto-inhibition, not auto-excitation (Qiu et al., 2016, Figure 2I). Therefore, optogenetic activation of AVPV Kiss1 neurons does not activate other AVPV Kiss1 neurons but appears to inhibit each other via GABA release. We have now added higher power confocal images (Figure 6A1-A3) illustrating that mCherry-labeled Kiss1^ARH^ neurons (Figure 6A1) project to the AVPV/PeN area but that the AVPV/PeN does not express mCherry-labeled cells (Figure 6A2), only immunoreactive kisspeptin neurons (Figure 6A3). Also, see Results section and Materials and methods section.

5) Indicate at what time after OVX or E2 replacement the experiments were performed as well as dose of E2 used.

The time of ovariectomy and E2 replacement and dose have been included in the Materials and methods section. Also, E2-treatments during the CPP have been added to Table 2.

6) Subsection “High frequency stimulation of glutamate release from Kiss1^ARH^ neurons excites POMC neurons and inhibits NPY neurons via metabotropic receptors”, "revealed a significantly reduced slow EPSP" However, Figure 11G does not show this different to be significant.

We have made it clear that we are comparing differences between the control and knockout with E2 replacement (Results section). (The response in E2-treated, OVX females; unpaired t-test, t _(25)_ = 2.735, p = 0.0113). a-a, p<0.05) in Figure 11G.

7) Has DHPC been used on NPY neurons and DCG-IV on POMC neurons?

Yes, and we have now included a statement in the Results section that DHPG had no effect on NPY, and DCG-IV had no effect on POMC neurons (see Results section).

8) The authors addressed the role of mGluR2 in NPY neurons, which is expressed only in 10% of them. Therefore, the study of mGluR7, which is more abundant, is missing.

We have now included the electrophysiological responses to mGluR7 agonists in oil and E2-treated OVX females (Results section; Figure 10D-F) and have also illustrated the effects of E2-treatment on mGluR7 mRNA expression in NPY neurons in Figure 10I.

9) E2 dramatically decreases Kiss1 expression (Figure 2) to suggest a decreased Kiss release. As such, it is expected that the Kisspeptin tone onto POMC and NPY neurons would be minimal at high Hz stimulation in OVX-E2 vs. OVX VGluT2 KO mice. Did the authors do these measurements? This should be at least discussed along with potential studies assessing the impact of E2 on responses to Kisspeptin in POMC or NPY neurons.

We did not compare E2-treated, OVX KO versus Oil-treated, OVX vGluT2 KO mice, since we had very few vGluT2 KO animals to work with. But we did compare E2-treated, OVX controls versus E2-treated, OVX vGluT2 KO mice. Please compare Figure 12I,J (KO) with Figure 7B,D (controls), animals in both groups were E2-treated. We have more fully emphasized the significant reduction in the postsynaptic responses in the Results section. In addition, we have discussed the possibility that kisspeptin excitatory action on POMC neurons and inhibitory actions on NPY/AgRP neurons could be one of the reasons why deleting vGluT2 was not effective to alter body weight on standard low fat chow over the time-course of our study (Discussion section).

10) Did the authors test the effect of high Hz stimulation of ARC Kiss1 on AVPV Kiss1 activity in VGluT2 KO mice? This aspect should be at least discussed.

Again, because of the limited number of vGluT2 KO animals, we did not systematically test the effect of high frequency stimulation on the input to the AVPV in vGlut2 KO. The AVPV/PeN Kiss1 neurons do not express Kiss1 or NKB receptors (Navarro et al., 2013). They do express κ receptors (Zhang et al., 2013) and would possibly be inhibited by dynorphin following high frequency stimulation of Kiss1^ARH^ inputs. We do not know if AVPV Kiss1 neurons express metabotropic glutamate receptors. We have added a note in the Discussion section that further studies are needed to determine whether glutamate and/or peptide neurotransmission from Kiss1^ARH^ to Kiss1^AVPV^ neurons are involved in reproductive functions.

[Editors' note: further revisions were requested prior to acceptance, as described below.]

The manuscript has been largely improved but there is one remaining issue that needs to be addressed before acceptance: could you please add a sentence justifying the use of the OVX and OVX+E2 animal models in the text?

We are pleased that our revised manuscript was favorably evaluated by the editors and reviewers, and we have made the final requested revision by including a sentence in the Materials and methods section justifying the use of OVX and OVX + E2 animal models. In addition, the Padilla et al., manuscript is published, and we have included the complete reference in the Reference list.